# Adversarial Training is a Form of Data-dependent Operator Norm Regularization

**Kevin Roth***
Dept of Computer Science
ETH Zürich
kevin.roth@inf.ethz.ch

**Yannic Kilcher***
Dept of Computer Science
ETH Zürich
yannic.kilcher@inf.ethz.ch

**Thomas Hofmann**
Dept of Computer Science
ETH Zürich
thomas.hofmann@inf.ethz.ch

## Abstract

We establish a theoretical link between adversarial training and operator norm regularization for deep neural networks. Specifically, we prove that $\ell_p$-norm constrained projected gradient ascent based adversarial training with an $\ell_q$-norm loss on the logits of clean and perturbed inputs is equivalent to data-dependent (p, q) operator norm regularization. This fundamental connection confirms the long-standing argument that a network's sensitivity to adversarial examples is tied to its spectral properties and hints at novel ways to robustify and defend against adversarial attacks. We provide extensive empirical evidence on state-of-the-art network architectures to support our theoretical results.

## 1 Introduction

While deep neural networks are known to be robust to random noise, it has been shown that their accuracy dramatically deteriorates in the face of so-called adversarial examples [4, 43, 17], i.e. small perturbations of the input signal, often imperceptible to humans, that are sufficient to induce large changes in the model output. This apparent vulnerability is worrisome as deep nets start to proliferate in the real-world, including in safety-critical deployments.

The most direct strategy of robustification, called adversarial training, aims to robustify a machine learning model by training it against an adversary that perturbs the examples before passing them to the model [17, 24, 30, 29, 26]. A different strategy of defense is to detect whether the input has been perturbed, by detecting characteristic regularities either in the adversarial perturbations themselves or in the network activations they induce [18, 14, 49, 27, 7, 38].

Despite practical advances in finding adversarial examples and defending against them, it is still an open question whether (i) adversarial examples are unavoidable, i.e. no robust model exists, cf. [11, 16], (ii) learning a robust model requires too much training data, cf. [40], (iii) learning a robust model from limited training data is possible but computationally intractable [6], or (iv) we just have not found the right model / training algorithm yet.

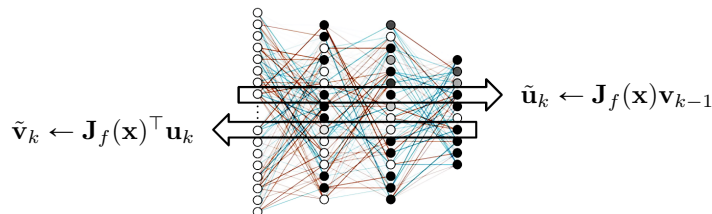

$$\tilde{\mathbf{v}}_k \leftarrow \mathbf{J}_f(\mathbf{x})^\top \mathbf{u}_k \qquad \tilde{\mathbf{u}}_k \leftarrow \mathbf{J}_f(\mathbf{x})\mathbf{v}_{k-1}$$

Figure 1: Our theoretical results suggest to think of iterative adversarial attacks as power-method-like forward-backward passes (indicated by the arrows) through (the Jacobian $\mathbf{J}_f(\mathbf{x})$ of) the network.

In this work, we investigate adversarial vulnerability in neural networks by focusing on the attack algorithms used to find adversarial examples. In particular, we make the following contributions:

- We present a data-dependent variant of spectral norm regularization that directly regularizes large singular values of a neural network in regions that are supported by the data, as opposed to existing methods that regularize a global, data-independent upper bound.

- We prove that $\ell_p$-norm constrained projected gradient ascent based adversarial training with an $\ell_q$-norm loss on the logits of clean and perturbed inputs is equivalent to data-dependent (p, q) operator norm regularization.

- We conduct extensive empirical evaluations showing among other things that (i) adversarial perturbations align with dominant singular vectors, (ii) adversarial training dampens the singular values, and (iii) adversarial training and data-dependent spectral norm regularization give rise to models that are significantly more linear around data than normally trained ones.

## 2  Related Work

The idea that a conservative measure of the sensitivity of a network to adversarial examples can be obtained by computing the spectral norm of the individual weight layers appeared already in the seminal work of Szegedy et al. [43]. A number of works have since suggested to regularize the global spectral norm [50, 28, 1, 10] and Lipschitz constant [8, 20, 46, 37] as a means to improve model robustness. Input gradient regularization has also been suggested [19, 25, 8]. Adversarial robustness has also been investigated via robustness to random noise [12, 13] and decision boundary tilting [44].

The most direct and popular strategy of robustification, however, is to use adversarial examples as data augmentation during training [17, 41, 39, 24, 35, 31, 29, 26]. Adversarial training can be viewed as a variant of (distributionally) robust optimization [9, 48, 2, 32, 42, 15] where a machine learning model is trained to minimize the worst-case loss against an adversary that can shift the entire training data within an uncertainty set. Interestingly, for certain problems and uncertainty sets, such as for linear regression and induced matrix norm balls, robust optimization has been shown to be equivalent to regularization [9, 48, 2, 3]. Similar results have been obtained also for (kernelized) SVMs [48].

We extend these lines of work by establishing a theoretical link between adversarial training and data-dependent operator norm regularization. This fundamental connection confirms the long-standing argument that a network's sensitivity to adversarial examples is tied to its spectral properties and opens the door for robust generalization bounds via data-dependent operator norm based ones.

## 3  Background

**Notation.** In this section we rederive global spectral norm regularization à la Yoshida & Miyato [50], while also setting up the notation for later. Let $\mathbf{x}$ and $y$ denote input-label pairs generated from a data distribution $P$. Let $f : \mathcal{X} \subset \mathbb{R}^n \to \mathbb{R}^d$ denote the logits of a $\theta$-parameterized piecewise linear classifier, i.e. $f(\cdot) = \mathbf{W}^L \phi^{L-1}(\mathbf{W}^{L-1}\phi^{L-2}(\dots) + \mathbf{b}^{L-1}) + \mathbf{b}^L$, where $\phi^\ell$ is the activation function, and $\mathbf{W}^\ell$, $\mathbf{b}^\ell$ denote the layer-wise weight matrix[1] and bias vector, collectively denoted by $\theta$. Let us furthermore assume that each activation function is a ReLU (the argument can easily be generalized to other piecewise linear activations). In this case, the activations $\phi^\ell$ act as input-dependent diagonal matrices $\Phi_{\mathbf{x}}^\ell := \mathrm{diag}(\phi_{\mathbf{x}}^\ell)$, where an element in the diagonal $\phi_{\mathbf{x}}^\ell := \mathbf{1}(\tilde{\mathbf{x}}^\ell \geqslant 0)$ is one if the corresponding pre-activation $\tilde{\mathbf{x}}^\ell := \mathbf{W}^\ell \phi^{\ell-1}(\cdot) + \mathbf{b}^\ell$ is positive and zero otherwise.

Following Raghu et al. [36], we call $\phi_{\mathbf{x}} := (\phi_{\mathbf{x}}^1, \dots, \phi_{\mathbf{x}}^{L-1}) \in \{0, 1\}^m$ the "activation pattern", where $m$ is the number of neurons in the network. For any activation pattern $\phi \in \{0, 1\}^m$ we can define the preimage $X(\phi) := \{\mathbf{x} \in \mathbb{R}^n : \phi_{\mathbf{x}} = \phi\}$, inducing a partitioning of the input space via $\mathbb{R}^n = \bigcup_\phi X(\phi)$. Note that some $X(\phi) = \varnothing$, as not all combinations of activiations may be feasible. See Figure 1 in [36] or Figure 3 in [33] for an illustration of ReLU tesselations of the input space.

**Linearization.** We can linearize $f$ within a neighborhood around $\mathbf{x}$ as follows

$$f(\mathbf{x} + \Delta\mathbf{x}) \simeq f(\mathbf{x}) + \mathbf{J}_f(\mathbf{x})\Delta\mathbf{x}, \quad (\text{with equality if } \mathbf{x} + \Delta\mathbf{x} \in X(\phi_{\mathbf{x}})), \qquad (1)$$

where $\mathbf{J}_f(\mathbf{x})$ denotes the Jacobian of $f$ at $\mathbf{x}$

$$\mathbf{J}_f(\mathbf{x}) = \mathbf{W}^L \cdot \Phi_{\mathbf{x}}^{L-1} \cdot \mathbf{W}^{L-1} \cdot \Phi_{\mathbf{x}}^{L-2} \cdots \Phi_{\mathbf{x}}^1 \cdot \mathbf{W}^1. \qquad (2)$$

For small $||\Delta\mathbf{x}||_2 \neq 0$, we have the following bound

$$\frac{||f(\mathbf{x}+\Delta\mathbf{x})-f(\mathbf{x})||_2}{||\Delta\mathbf{x}||_2} \simeq \frac{||\mathbf{J}_f(\mathbf{x})\Delta\mathbf{x}||_2}{||\Delta\mathbf{x}||_2} \leqslant \sigma(\mathbf{J}_f(\mathbf{x})) := \max_{\mathbf{v}:||\mathbf{v}||_2=1} ||\mathbf{J}_f(\mathbf{x})\mathbf{v}||_2 \qquad (3)$$

where $\sigma(\mathbf{J}_f(\mathbf{x}))$ is the *spectral norm* (largest singular value) of the linear operator $\mathbf{J}_f(\mathbf{x})$. From a robustness perspective we want $\sigma(\mathbf{J}_f(\mathbf{x}))$ to be small in regions that are supported by the data.

**Global Spectral Norm Regularization.** Based on the factorization in Eq. 2 and the non-expansiveness of the activations, $\sigma(\Phi_\mathbf{x}^\ell) \leqslant 1, \forall\ell \in \{1,...,L-1\}$, Yoshida & Miyato [50] suggested to upper-bound the spectral norm of the Jacobian by the *data-independent*(!) product of the spectral norms of the weight matrices $\sigma(\mathbf{J}_f(\mathbf{x})) \leqslant \prod_{\ell=1}^{L} \sigma(\mathbf{W}^\ell), \forall\mathbf{x} \in \mathcal{X}$. The layer-wise spectral norms $\sigma^\ell := \sigma(\mathbf{W}^\ell)$ can be computed iteratively using the power method[2]. Starting from a random $\mathbf{v}_0$,

$$\mathbf{u}_k^\ell \leftarrow \tilde{\mathbf{u}}_k^\ell/||\tilde{\mathbf{u}}_k^\ell||_2 \ , \ \tilde{\mathbf{u}}_k^\ell \leftarrow \mathbf{W}^\ell \mathbf{v}_{k-1}^\ell \ , \quad \mathbf{v}_k^\ell \leftarrow \tilde{\mathbf{v}}_k^\ell/||\tilde{\mathbf{v}}_k^\ell||_2 \ , \ \tilde{\mathbf{v}}_k^\ell \leftarrow (\mathbf{W}^\ell)^\top \mathbf{u}_k^\ell \ . \qquad (4)$$

The (final) singular value can be computed as $\sigma_k^\ell = (\mathbf{u}_k^\ell)^\top \mathbf{W}^\ell \mathbf{v}_k^\ell$.

Yoshida & Miyato [50] suggest to turn this upper-bound into a global (data-independent) regularizer

$$\min\theta \rightarrow \mathbf{E}_{(\mathbf{x},y)\sim\hat{P}}\left[\ell(y,f(\mathbf{x}))\right] + \frac{\lambda}{2}\sum_{\ell=1}^{L}\sigma(\mathbf{W}^\ell)^2 \ , \qquad (5)$$

where $\ell(\cdot,\cdot)$ denotes a classification loss. It can be verified that $\nabla_\mathbf{W}\sigma(\mathbf{W})^2/2 = \sigma\mathbf{u}\mathbf{v}^\top$, with $\sigma, \mathbf{u}, \mathbf{v}$ being the principal singular value/vectors of $\mathbf{W}$. Eq. 5 thus effectively adds a term $\lambda\sigma^\ell\mathbf{u}^\ell(\mathbf{v}^\ell)^\top$ to the parameter gradient of each layer $\ell$. In terms of computational complexity, because the global regularizer decouples from the empirical loss, the power-method can be amortized across data-points, hence a single power method iteration per parameter update step usually suffices in practice [50].

**Global vs. Local Regularization** Global bounds trivially generalize from the training to the test set. The problem however is that they can be arbitrarily loose, e.g. penalizing the spectral norm over irrelevant regions of the ambient space. To illustrate this, consider the ideal robust classifier that is essentially piecewise constant on class-conditional regions, with sharp transitions between the classes. The global spectral norm will be heavily influenced by the sharp transition zones, whereas a local data-dependent bound can adapt to regions where the classifier is approximately constant [20]. We would therefore expect a global regularizer to have the largest effect in the empty parts of the input space. A local regularizer, on the contrary, has its main effect around the data manifold.

## 4 Adversarial Training is a Form of Operator Norm Regularization

### 4.1 Data-dependent Spectral Norm Regularization

We now show how to directly regularize the *data-dependent* spectral norm of the Jacobian $\mathbf{J}_f(\mathbf{x})$. Assuming that the dominant singular value is non-degenerate[2], the largest singular value and the corresponding left and right singular vectors can efficiently be computed via the power method. Starting from $\mathbf{v}_0$, we successively compute the unnormalized (denoted with a tilde) and normalized approximations to the dominant singular vectors,

$$\begin{aligned}\mathbf{u}_k &\leftarrow \tilde{\mathbf{u}}_k/||\tilde{\mathbf{u}}_k||_2 \ , \ \tilde{\mathbf{u}}_k \leftarrow \mathbf{J}_f(\mathbf{x})\mathbf{v}_{k-1} \\ \mathbf{v}_k &\leftarrow \tilde{\mathbf{v}}_k/||\tilde{\mathbf{v}}_k||_2 \ , \ \tilde{\mathbf{v}}_k \leftarrow \mathbf{J}_f(\mathbf{x})^\top \mathbf{u}_k \ . \end{aligned} \qquad (6)$$

The (final) singular value can then be computed via $\sigma(\mathbf{J}_f(\mathbf{x})) = \mathbf{u}^\top\mathbf{J}_f(\mathbf{x})\mathbf{v}$. For brevity we suppress the dependence of $\mathbf{u}, \mathbf{v}$ on $\mathbf{x}$ in the rest of the paper. Note that $\mathbf{v}$ gives the direction in input space that corresponds to the steepest ascent of the linearized network along $\mathbf{u}$. See Figure 1 for an illustration of the forward-backward passes through ($\mathbf{J}_f(\mathbf{x})$ of) the network.

We can turn this into a regularizer by learning the parameters $\theta$ via

$$\min\theta \rightarrow \mathbf{E}_{(\mathbf{x},y)\sim\hat{P}}\left[\ell(y,f(\mathbf{x})) + \frac{\tilde{\lambda}}{2}\sigma(\mathbf{J}_f(\mathbf{x}))^2\right] , \qquad (7)$$

where the data-dependent singular value $\sigma(\mathbf{J}_f(\mathbf{x}))$ is computed via Eq. 6.

By optimality / stationarity[3] we also have that $\sigma(\mathbf{J}_f(\mathbf{x})) = \mathbf{u}^\top \mathbf{J}_f(\mathbf{x})\mathbf{v} = ||\mathbf{J}_f(\mathbf{x})\mathbf{v}||_2$. Totherther with linearization $\epsilon \mathbf{J}_f(\mathbf{x})\mathbf{v} \simeq f(\mathbf{x} + \epsilon\mathbf{v}) - f(\mathbf{x})$ (which holds with equality if $\mathbf{x} + \epsilon\mathbf{v} \in X(\phi_\mathbf{x})$), we can regularize learning also via a sum-of-squares based spectral norm regularizer $\frac{\tilde{\lambda}}{2}||f(\mathbf{x}+\epsilon\mathbf{v}) - f(\mathbf{x})||_2^2$, where $\tilde{\lambda} = \lambda\epsilon^2$. Both variants can readily be implemented in modern deep learning frameworks. We found the sum-of-squares based one to be slightly more numerically stable.

In terms of computational complexity, the data-dependent regularizer is equally expensive as projected gradient ascent based adversarial training, and both are a constant (number of power method iterations) times more expensive than the data-independent variant, plus an overhead that depends on the batch size, which is mitigated in modern frameworks by parallelizing computations across a batch of data.

### 4.2 Data-dependent Operator Norm Regularization

More generally, we can directly regularize the data-dependent $(p, q)$-operator norm of the Jacobian. To this end, let the input-space (domain) and output-space (co-domain) of the linear map $\mathbf{J}_f(\mathbf{x})$ be equipped with the $\ell_p$ and $\ell_q$ norms respectively, defined as $||\mathbf{x}||_p := (\sum_{i=1}^n |x_i|^p)^{1/p}$ for $1 \leqslant p < \infty$ and $||\mathbf{x}||_\infty := \max_i |x_i|$ for $p = \infty$. The *data-dependent $(p, q)$-operator norm* of $\mathbf{J}_f(\mathbf{x})$ is defined as

$$||\mathbf{J}_f(\mathbf{x})||_{p,q} := \max_{\mathbf{v}:||\mathbf{v}||_p=1} ||\mathbf{J}_f(\mathbf{x})\mathbf{v}||_q \tag{8}$$

which is a data-dependent measure of the maximal amount of signal-gain that can be induced when propagating a norm-bounded input vector through the linearized network. Table 1 shows $(p, q)$-operator norms for typical values of $p$ (domain) and $q$ (co-domain).

For general $(p, q)$-norms, the maximizer $\mathbf{v}$ in Eq. 8 can be computed via projected gradient ascent. Let $\mathbf{v}_0$ be a random vector or an approximation to the maximizer. The data-dependent $(p, q)$-operator norm of $\mathbf{J}_f(\mathbf{x})$ can be computed iteratively via

$$\mathbf{v}_k = \Pi_{\{||\cdot||_p=1\}}(\mathbf{v}_{k-1} + \alpha \nabla_\mathbf{v} ||\mathbf{J}_f(\mathbf{x})\mathbf{v}_{k-1}||_q) \tag{9}$$

where $\Pi_{\{||\cdot||_p=1\}}(\tilde{\mathbf{v}}) := \arg\min_{\mathbf{v}*:||\mathbf{v}*||_p=1} ||\mathbf{v}^* - \tilde{\mathbf{v}}||_2$ is the orthogonal projection onto the $\ell_p$ unit sphere, and where $\alpha$ is a step-size or weighting factor, trading off the previous iterate $\mathbf{v}_{k-1}$ with the current gradient step $\nabla_\mathbf{v} ||\mathbf{J}_f(\mathbf{x})\mathbf{v}_{k-1}||_q$.

By the chain-rule, the computation of the gradient step

$$\nabla_\mathbf{v} ||\mathbf{J}_f(\mathbf{x})\mathbf{v}||_q = \mathbf{J}_f(\mathbf{x})^\top \text{sign}(\mathbf{u}) \odot |\mathbf{u}|^{q-1}/||\mathbf{u}||_q^{q-1} \;, \quad \text{where } \mathbf{u} = \mathbf{J}_f(\mathbf{x})\mathbf{v}\,, \tag{10}$$

can be decomposed into a forward and backward pass through the Jacobian $\mathbf{J}_f(\mathbf{x})$, yielding the following *projected gradient ascent based operator norm iteration method*

$$\begin{aligned}
\mathbf{u}_k &\leftarrow \text{sign}(\tilde{\mathbf{u}}_k) \odot |\tilde{\mathbf{u}}_k|^{q-1}/||\tilde{\mathbf{u}}_k||_q^{q-1}\,, \quad \tilde{\mathbf{u}}_k \leftarrow \mathbf{J}_f(\mathbf{x})\mathbf{v}_{k-1} \\
\mathbf{v}_k &\leftarrow \Pi_{\{||\cdot||_p=1\}}(\mathbf{v}_{k-1} + \alpha\,\tilde{\mathbf{v}}_k)\,, \qquad \tilde{\mathbf{v}}_k \leftarrow \mathbf{J}_f(\mathbf{x})^\top \mathbf{u}_k
\end{aligned} \tag{11}$$

where $\odot$, $\text{sign}(\cdot)$ and $|\cdot|$ denote elementwise product, sign and absolute-value. See Figure 1 for an illustration of the forward-backward passes through (the Jacobian $\mathbf{J}_f(\mathbf{x})$ of) the network.

In particular, we have the following well-known special cases. For $q = 2$, the forward pass equation is given by $\mathbf{u}_k \leftarrow \tilde{\mathbf{u}}_k/||\tilde{\mathbf{u}}_k||_2$, while for $q = 1$ it is given by $\mathbf{u}_k \leftarrow \text{sign}(\tilde{\mathbf{u}}_k)$. The $q = \infty$ limit on the other hand is given by $\lim_{q\to\infty} \text{sign}(\tilde{\mathbf{u}}) \odot |\tilde{\mathbf{u}}|^{q-1}/||\tilde{\mathbf{u}}||_q^{q-1} = |\mathcal{I}|^{-1}\text{sign}(\tilde{\mathbf{u}}) \odot \mathbf{1}_\mathcal{I}$ with $\mathbf{1}_\mathcal{I} = \sum_{i\in\mathcal{I}} \mathbf{e}_i$, where $\mathcal{I} := \{j \in [1, ..., d] : |\tilde{u}_j| = ||\tilde{\mathbf{u}}||_\infty\}$ denotes the set of indices at which $\tilde{\mathbf{u}}$ attains its maximum norm and $\mathbf{e}_i$ is the $i$-th canonical unit vector. See Sec. 7.5 in the Appendix for a derivation.

It is interesting to note that we can recover the power method update equations by taking the limit $\alpha \to \infty$ (which is well-defined since $\alpha$ is inside the projection) in the above iteration equations, which we consider to be an interesting result in its own right, see **Lemma 2** in the Appendix. With this, we obtain the following *power method limit* of the operator norm iteration equations

$$\begin{aligned}
\mathbf{u}_k &\leftarrow \text{sign}(\tilde{\mathbf{u}}_k) \odot |\tilde{\mathbf{u}}_k|^{q-1}/||\tilde{\mathbf{u}}_k||_q^{q-1}\,, \quad \tilde{\mathbf{u}}_k \leftarrow \mathbf{J}_f(\mathbf{x})\mathbf{v}_{k-1} \\
\mathbf{v}_k &\leftarrow \text{sign}(\tilde{\mathbf{v}}_k) \odot |\tilde{\mathbf{v}}_k|^{p*-1}/||\tilde{\mathbf{v}}_k||_{p*}^{p*-1}\,, \quad \tilde{\mathbf{v}}_k \leftarrow \mathbf{J}_f(\mathbf{x})^\top \mathbf{u}_k
\end{aligned} \tag{12}$$

The condition that $\alpha \to \infty$ means that in the update equation for $\mathbf{v}_k$ all the weight is placed on the current gradient direction $\tilde{\mathbf{v}}_k$ whereas no weight is put on the previous iterate $\mathbf{v}_{k-1}$. See [5, 21] for a convergence analysis of the power method.

We can turn this into a regularizer by learning the parameters $\theta$ via

$$\min \theta \to \mathbf{E}_{(\mathbf{x},y) \sim \hat{P}}\Big[\ell(y, f(\mathbf{x})) + \tilde{\lambda} \, ||\mathbf{J}_f(\mathbf{x})||_{p,q}\Big] \tag{13}$$

Note that we can also use the $q$-th power of the operator norm as a regularizer (with a prefactor of $1/q$). It is easy to see that this only affects the normalization of $\mathbf{u}_k$.

### 4.3 Power Method Formulation of Adversarial Training

Adversarial training [17, 24, 26] aims to improve the robustness of a model by training it against an adversary that perturbs each training example subject to a proximity constraint, e.g. in $\ell_p$-norm,

$$\min \theta \to \mathbf{E}_{(\mathbf{x},y) \sim \hat{P}}\Big[\ell(y, f(\mathbf{x})) + \lambda \max_{\mathbf{x}* \in \mathcal{B}_\epsilon^p(\mathbf{x})} \ell_{\mathrm{adv}}(y, f(\mathbf{x}*))\Big] \tag{14}$$

where $\ell_{\mathrm{adv}}(\cdot, \cdot)$ denotes the loss function used to find adversarial perturbations (not necessarily the same as the classification loss $\ell(\cdot, \cdot)$).

The adversarial example $\mathbf{x}*$ is typically computed iteratively, e.g. via $\ell_p$-norm constrained projected gradient ascent (cf. the PGD attack in [34] or Sec. 3 on PGD-based adversarial attacks in [47])

$$\mathbf{x}_0 \sim \mathcal{U}(\mathcal{B}_\epsilon^p(\mathbf{x})), \quad \mathbf{x}_k = \Pi_{\mathcal{B}_\epsilon^p(\mathbf{x})}\Big(\mathbf{x}_{k-1} + \alpha \arg\max_{\mathbf{v}_k : ||\mathbf{v}_k||_p \leqslant 1} \mathbf{v}_k^\top \nabla_{\mathbf{x}}\ell_{\mathrm{adv}}(y, f(\mathbf{x}_{k-1}))\Big) \tag{15}$$

where $\Pi_{\mathcal{B}_\epsilon^p(\mathbf{x})}(\tilde{\mathbf{x}}) := \arg\min_{\mathbf{x}* \in \mathcal{B}_\epsilon^p(\mathbf{x})} ||\mathbf{x}* - \tilde{\mathbf{x}}||_2$ is the orthogonal projection operator into the norm ball $\mathcal{B}_\epsilon^p(\mathbf{x}) := \{\mathbf{x}* : ||\mathbf{x}* - \mathbf{x}||_p \leqslant \epsilon\}$, $\alpha$ is a step-size or weighting factor, trading off the previous iterate $\mathbf{x}_{k-1}$ with the current gradient step $\mathbf{v}_k$, and $y$ is the true or predicted label. For targeted attacks the sign in front of $\alpha$ is flipped, so as to descend the loss function into the direction of the target label.

We can in fact derive the following explicit expression for the optimal perturbation to a linear function under an $\ell_p$-norm constraint, see **Lemma 1** in the Appendix,

$$\mathbf{v}* = \arg\max_{\mathbf{v} : ||\mathbf{v}||_p \leqslant 1} \mathbf{v}^\top \mathbf{z} = \mathrm{sign}(\mathbf{z}) \odot |\mathbf{z}|^{p*-1}/||\mathbf{z}||_{p*}^{p*-1} \tag{16}$$

where $\odot$, $\mathrm{sign}(\cdot)$ and $|\cdot|$ denote elementwise product, sign and absolute-value, $p*$ is the Hölder conjugate of $p$, given by $1/p + 1/p* = 1$, and $\mathbf{z}$ is an arbitrary non-zero vector, e.g. $\mathbf{z} = \nabla_{\mathbf{x}}\ell_{\mathrm{adv}}(y, f(\mathbf{x}))$. The derivation can be found in Sec. 7.6 in the Appendix.

As a result, $\ell_p$-norm constrained projected gradient ascent can be implemented as follows

$$\mathbf{x}_k = \Pi_{\mathcal{B}_\epsilon^p(\mathbf{x})}\Big(\mathbf{x}_{k-1} + \alpha \, \mathrm{sign}(\nabla_{\mathbf{x}}\ell_{\mathrm{adv}}) \odot |\nabla_{\mathbf{x}}\ell_{\mathrm{adv}}|^{p*-1}/||\nabla_{\mathbf{x}}\ell_{\mathrm{adv}}||_{p*}^{p*-1}\Big) \tag{17}$$

where $p*$ is given by $1/p + 1/p* = 1$ and $\nabla_{\mathbf{x}}\ell_{\mathrm{adv}}$ is short-hand notation for $\nabla_{\mathbf{x}}\ell_{\mathrm{adv}}(y, f(\mathbf{x}_{k-1}))$.

By the chain-rule, the computation of the gradient-step $\tilde{\mathbf{v}}_k := \nabla_{\mathbf{x}}\ell_{\mathrm{adv}}(y, f(\mathbf{x}_{k-1}))$ can be decomposed into a logit-gradient and a Jacobian vector product. $\ell_p$-norm constrained projected gradient ascent can thus equivalently be written in the following *power method like* forward-backward pass form (the normalization of $\tilde{\mathbf{u}}_k$ is optional and can be absorbed into the normalization of $\tilde{\mathbf{v}}_k$)

$$\mathbf{u}_k \leftarrow \tilde{\mathbf{u}}_k/||\tilde{\mathbf{u}}_k||_2, \quad \tilde{\mathbf{u}}_k \leftarrow \nabla_{\mathbf{z}}\ell_{\mathrm{adv}}(y, \mathbf{z})|_{\mathbf{z}=f(\mathbf{x}_{k-1})}$$
$$\mathbf{v}_k \leftarrow \mathrm{sign}(\tilde{\mathbf{v}}_k) \odot |\tilde{\mathbf{v}}_k|^{p*-1}/||\tilde{\mathbf{v}}_k||_{p*}^{p*-1}, \quad \tilde{\mathbf{v}}_k \leftarrow \mathbf{J}_f(\mathbf{x}_{k-1})^\top \mathbf{u}_k \tag{18}$$
$$\mathbf{x}_k \leftarrow \Pi_{\mathcal{B}_\epsilon^p(\mathbf{x})}(\mathbf{x}_{k-1} + \alpha \mathbf{v}_k)$$

The adversarial loss function determines the logit-space direction $\mathbf{u}_k$ in the power method like formulation of adversarial training, while $\tilde{\mathbf{v}}_k$ resp. $\mathbf{v}_k$ gives the unconstrained resp. norm-constrained direction in input space that corresponds to the steepest ascent of the linearized network along $\mathbf{u}_k$.

The corresponding forward-backward pass equations for an $\ell_q$-norm loss on the logits of the clean and perturbed input $\ell_{\mathrm{adv}}(f(\mathbf{x}), f(\mathbf{x}*)) = ||f(\mathbf{x}) - f(\mathbf{x}*)||_q$ are shown in Eq. 108 in the Appendix.

Table 1: Computing $(p,q)$-operator norms for typical values of $p$ (domain) and $q$ (co-domain). See Sec. 4.3.1 in [45]. We prove Theorem 1 for all the entries in this table. In our setting, "columns" of $\mathbf{J}_f(\mathbf{x})$ correspond to paths through the network originating in a specific input neuron, whereas "rows" correspond to paths ending in a specific output neuron.

|  |  | Co-domain | | |
|---|---|---|---|---|
|  |  | $\ell_1$ | $\ell_2$ | $\ell_\infty$ |
| Domain | $\ell_1$ | max $\ell_1$ norm of a column | max $\ell_2$ norm of a column | max $\ell_\infty$ norm of a column |
|  | $\ell_2$ | NP-hard | max singular value | max $\ell_2$ norm of a row |
|  | $\ell_\infty$ | NP-hard | NP-hard | max $\ell_1$ norm of a row |

Comparing Eq. 18 with Eq. 12, we can see that *adversarial training is a form of data-dependent operator norm regularization*. The following theorem states the precise conditions under which they are mathematically equivalent. The correspondence is proven for $\ell_q$-norm adversarial losses [39, 22]. See Sec. 7.4 in the Appendix for a discussion of the softmax cross-entropy loss. The proof can be found in Sec. 7.8 in the Appendix.

> **Theorem 1.** *For $\epsilon$ small enough such that $\mathcal{B}_\epsilon^p(\mathbf{x}) \subset X(\phi_{\mathbf{x}})$ and in the limit $\alpha \to \infty$, $\ell_p$-norm constrained projected gradient ascent based adversarial training with an $\ell_q$-norm loss on the logits of the clean and perturbed input $\ell_{\mathrm{adv}}(f(\mathbf{x}), f(\mathbf{x}^*)) = ||f(\mathbf{x}) - f(\mathbf{x}^*)||_q$, with $p, q \in \{1, 2, \infty\}$, is equivalent to the power method limit of data-dependent $(p, q)$-operator norm regularization of the Jacobian $\mathbf{J}_f(\mathbf{x})$ of the network.*

In practice, the correspondence holds *approximately* (to a very good degree) in a region much larger than $X(\phi_{\mathbf{x}})$, namely as long as the Jacobian of the network remains approximately constant in the uncertainty ball under consideration, see Sec. 5.4, specifically Figure 4 (left), as well as Sec. 5.5, specifically Figure 5 and Figure 9.

**In summary**, our Theorem confirms that a network's sensitivity to adversarial examples is characterized through its spectral properties: it is the dominant singular vector (resp. the maximizer $\mathbf{v}^*$ in Lemma 1) corresponding to the largest singular value (resp. the $(p, q)$-operator norm) that determines the optimal adversarial perturbation and hence the sensitivity of the model to adversarial examples.

Our results also explain why input gradient regularization and fast gradient method based adversarial training do not sufficiently protect against iterative adversarial attacks, namely because the input gradient, resp. a single power method iteration, do not yield a sufficiently good approximation for the dominant singular vector in general. Similarly, we do not expect Frobenius norm (= sum of all singular values) regularization to work as well as data-dependent spectral norm (= largest s.v.) regularization in robustifying against iterative adversarial attacks. More details in Sec. 7.2 & 7.3.

# 5 Experimental Results

## 5.1 Dataset, Architecture & Training Methods

We trained Convolutional Neural Networks (CNNs) with batch normalization on the CIFAR10 data set [23]. We use a 7-layer CNN as our default platform, since it has good test set accuracy at acceptable computational requirements. For the robustness experiments, we also train a Wide Residual Network (WRN-28-10) [51]. We used an estimated 6k TitanX GPU hours in total for all our experiments.

We train each classifier with a number of different training methods: (i) 'Standard': standard empirical risk minimization with a softmax cross-entropy loss, (ii) 'Adversarial': $\ell_2$- / $\ell_\infty$-norm constrained projected gradient ascent (PGA) based adversarial training, (iii) 'global SNR': global spectral norm regularization à la Yoshida & Miyato [50], and (iv) 'd.d. SNR / ONR': data-dependent spectral / operator norm regularization.

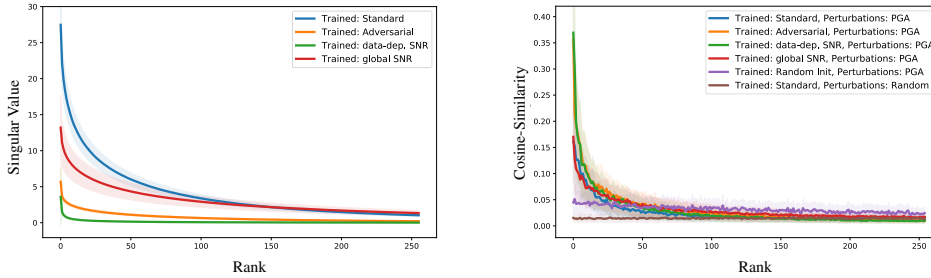

Figure 2: (Left) Singular value spectrum of the Jacobian $\mathbf{J}_{\phi^{L-1}}(\mathbf{x})$ for networks $f = \mathbf{W}^L \phi^{L-1}$ trained with different training methods. (Right) Alignment of adversarial perturbations with singular vectors $\mathbf{v}_r$ of the Jacobian $\mathbf{J}_{\phi^{L-1}}(\mathbf{x})$, as a function of the rank $r$ of the singular vector. For comparison we also show the cosine-similarity with the singular vectors of a random network. We can see that (i) adversarial training and data-dependent spectral norm regularization significantly dampen the singular values, while global spectral norm regularization has almost no effect compared to standard training, and (ii) adversarial perturbations are strongly aligned with dominant singular vectors.

As a default attack strategy we use an $\ell_2$- / $\ell_\infty$-norm constrained PGA white-box attack with 10 attack iterations. We verified that all our conclusions also hold for larger numbers of attack iterations, however, due to computational constraints we limit to 10. The attack strength $\epsilon$ used for training (indicated by a vertical dashed line in the Figures below) was chosen to be the smallest value such that almost all adversarially perturbed inputs to the standard model are successfully misclassified.

The regularization constants were chosen such that the regularized models achieve the same test set accuracy on clean examples as the adversarially trained model does. Global SNR is implemented with one spectral norm update per training step, as recommended by the authors. We provide additional experiments for global SNR with 10 update iterations in Sec. 7.16 in the Appendix. As shown, the 10 iterations make no difference, in line with the regularizer's decoupling from the empirical loss.

Table 2 in the Appendix summarizes the test set accuracies for the training methods we considered. Additional experimental results and further details regarding the experimental setup can be found in Secs. 7.10 and following in the Appendix.

Shaded areas in the plots below denote standard errors w.r.t. the number of test set samples over which the experiment was repeated.

## 5.2 Spectral Properties

**Effect of training method on singular value spectrum.** We compute the singular value spectrum of the Jacobian $\mathbf{J}_{\phi^{L-1}}(\mathbf{x})$ for networks $f = \mathbf{W}\phi^{L-1}$ trained with different training methods and evaluated at a number of different test examples. Since we are interested in computing the full singular value spectrum, and not just the dominant singular value / vectors as during training, using the power method would be too impractical, as it gives us access to only one (the dominant) singular value-vector pair at a time. Instead, we first extract the Jacobian (which is *per se* defined as a computational graph in modern deep learning frameworks) as an input-dim×output-dim matrix and then use available matrix factorization routines to compute the full SVD of the extracted matrix. For each training method, the procedure is repeated for 200 randomly chosen clean and corresponding adversarially perturbed test examples. Further details regarding the Jacobian extraction can be found in Sec. 7.9 in the Appendix. The results are shown in Figure 2 (left). We can see that adversarial training and data-dependent spectral norm regularization significantly dampen the singular values, while global spectral norm regularization has almost no effect compared to standard training.

**Alignment of adversarial perturbations with singular vectors.** We compute the cosine-similarity of adversarial perturbations with singular vectors $\mathbf{v}_r$ of the Jacobian $\mathbf{J}_{\phi^{L-1}}(\mathbf{x})$, extracted at a number of test set examples, as a function of the rank of the singular vectors returned by the SVD decomposition. For comparison we also show the cosine-similarity with the singular vectors of a random network. The results are shown in Figure 2 (right). We can see that for all training methods (except the random network) adversarial perturbations are strongly aligned with the dominant singular vectors while the alignment decreases with increasing rank.

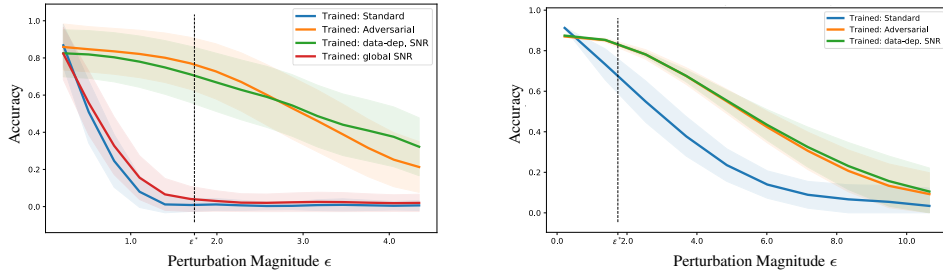

Figure 3: Classification accuracy as a function of perturbation strength $\epsilon$. (Left) 7-layer CNN (Right) WideResNet WRN-28-10. The dashed line indicates the $\epsilon$ used during training. Curves were aggregated over 2000 (left) resp. all (right) samples from the test set. We can see that adversarially trained and data-dependent spectral norm regularized models are equally robust to adversarial attacks.

Interestingly, this strong alignment with dominant singular vectors also confirms why input gradient regularization and fast gradient method (FGM) based adversarial training do not sufficiently protect against iterative adversarial attacks, namely because the input gradient, resp. a single power method iteration, do not yield a sufficiently good approximation for the dominant singular vector in general. See Sec. 7.2 in the Appendix for a more technical explanation.

### 5.3 Adversarial Robustness

**Adversarial classification accuracy.** A plot of the classification accuracy on adversarially perturbed test examples, as a function of the perturbation strength $\epsilon$, is shown in Figure 3. We can see that data-dependent spectral norm regularized models are equally robust to adversarial examples as adversarially trained models and both are significantly more robust than a normally trained one, while global spectral norm regularization does not seem to robustify the model substantially. This is in line with our earlier observation that adversarial perturbations tend to align with dominant singular vectors and that they are dampened by adversarial training and data-dependent spectral norm regularization.

**Interpolating between AT and d.d. SNR.** We have also conducted an experiment where we train several networks from scratch each with an objective function that convexly combines adversarial training with data-dependent spectral norm regularization in a way that allows us to interpolate between (i) the fraction of adversarial examples relative to clean examples used during adversarial training controlled by $\lambda$ in Eq. 14 and (ii) the regularization parameter $\tilde{\lambda}$ in Eq. 13. This allows us to continuously trade-off the contribution of AT with that of d.d. SNR in the empirical risk minimization. The results, shown in Figure 7, again confirm that the two training methods are equivalent.

### 5.4 Local Linearity

**Validity of linear approximation.** To determine the range in which the locally linear approximation is valid, we measure the deviation from linearity $||\phi^{L-1}(\mathbf{x} + \mathbf{z}) - (\phi^{L-1}(\mathbf{x}) + \mathbf{J}_{\phi^{L-1}}(\mathbf{x})\mathbf{z})||_2$ as the distance $||\mathbf{z}||_2$ is increased in random and adversarial directions $\mathbf{z}$, with adversarial perturbations serving as a proxy for the direction in which the linear approximation holds the least. This allows us to investigate how good the linear approximation for different training methods is, as an increasing number of activation boundaries are crossed with increasing perturbation radius.

The results are shown in Figure 4 (left). We can see that adversarially trained and data-dependent spectral norm regularized models are significantly more linear than the normally trained one and that the Jacobian $\mathbf{J}_f(\mathbf{x})$ is a good approximation for the AT and d.-d. SNR regularized classifier in the entire $\epsilon^*$-ball around $\mathbf{x}$, since the models remain flat even in the adversarial direction for perturbation magnitudes up to the order of the $\epsilon^*$ used during adversarial training (dashed vertical line). Moreover, since our Theorem is applicable as long as the Jacobian of the network remains (approximately) constant in the uncertainty ball under consideration, this also means that the correspondence between AT and d.d. SNR holds up to the size of the $\epsilon^*$-ball commonly used in AT practice.

**Largest singular value over distance.** Figure 4 (right) shows the largest singular value of the linear operator $\mathbf{J}_{\phi^{L-1}}(\mathbf{x}+\mathbf{z})$ as the distance $||\mathbf{z}||_2$ from $\mathbf{x}$ is increased, both along random and adversarial directions $\mathbf{z}$. We can see that the naturally trained network develops large dominant singular values around the data point, while the adversarially trained and data-dependent spectral norm regularized models manage to keep the dominant singular value low in the vicinity of $\mathbf{x}$.

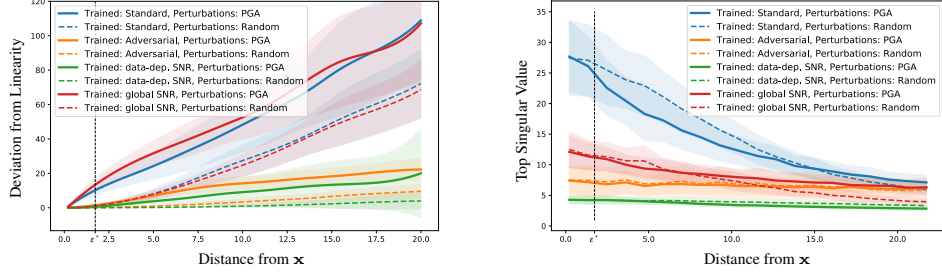

Figure 4: (Left) Deviation from linearity $||\phi^{L-1}(\mathbf{x} + \mathbf{z}) - (\phi^{L-1}(\mathbf{x}) + \mathbf{J}_{\phi^{L-1}}(\mathbf{x})\mathbf{z})||_2$ as a function of the distance $||\mathbf{z}||_2$ from $\mathbf{x}$ for random and adversarial perturbations $\mathbf{z}$. (Right) Largest singular value of the Jacobian $\mathbf{J}_{\phi^{L-1}}(\mathbf{x}+\mathbf{z})$ as a function of the magnitude $||\mathbf{z}||_2$. We can see that adversarially trained and data-dependent spectral norm regularized models are significantly more linear around data points than the normally trained one and that the Jacobian $\mathbf{J}_f(\mathbf{x})$ is a good approximation for the AT and d.-d. SNR regularized classifier in the entire $\epsilon^*$-ball around $\mathbf{x}$.

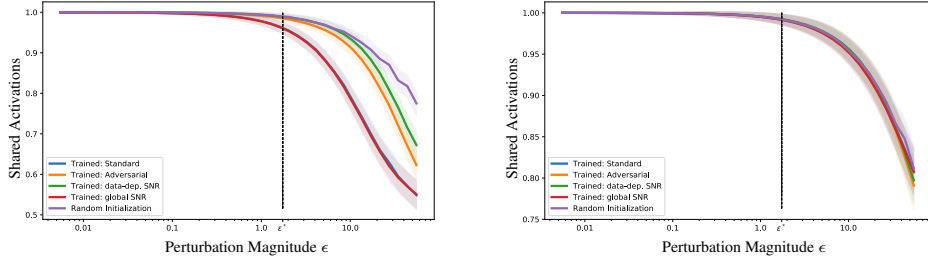

Figure 5: Fraction of shared activations as a function of noise magnitude $\epsilon$ between activation patterns $\phi_{(\cdot)}$ and $\phi_{(\cdot)+\mathbf{z}}$, where $(\cdot)$ is either (left) a data point $\mathbf{x}$ sampled from the test set, or (right) a data point $\mathbf{n}$ sampled uniformly from the input domain (a.s. not on the data manifold) and $\mathbf{z}$ is uniform noise of magnitude $\epsilon$. We can see that both d.d. SNR and AT significantly increase the size of the ReLU cells around data, yet have no effect away from the data manifold.

## 5.5 Activation patterns

An important property of *data-dependent* regularization is that it primarily acts on the data manifold whereas it should have comparatively little effect on irrelevant parts of the input space, see Sec. 3. We test this hypothesis by comparing activation patterns $\phi_{(\cdot)} \in \{0, 1\}^m$ of perturbed input samples. We measure the fraction of shared activations $m^{-1}(\phi_{(\cdot)} \cap \phi_{(\cdot)+\mathbf{z}})$, where $(\cdot)$ is either a data point $\mathbf{x}$ sampled from the test set, shown in Figure 5 (left), or a data point $\mathbf{n}$ sampled uniformly from the input domain (a.s. not on the data manifold), shown in Figure 5 (right), and where $\mathbf{z}$ is a random uniform noise vector of magnitude $\epsilon$. From these curves, we can estimate the average size of the ReLU cells making up the activation pattern $\phi_{(\cdot)}$. we can see that both d.d. SNR and AT significantly increase the size of the ReLU cells around data (in both random and adv. directions), thus improving the stability of activation patterns against adversarial examples, yet they have no effect away from the data manifold. See also Figure 9 in the Appendix.

## 6 Conclusion

We established a theoretical link between adversarial training and operator norm regularization for deep neural networks. Specifically, we derive the precise conditions under which $\ell_p$-norm constrained projected gradient ascent based adversarial training with an $\ell_q$-norm loss on the logits of clean and perturbed inputs is equivalent to data-dependent (p, q) operator norm regularization. This fundamental connection confirms the long-standing argument that a network's sensitivity to adversarial examples is tied to its spectral properties. We also conducted extensive empirical evaluations confirming the theoretically predicted effect of adversarial training and data-dependent operator norm regularization on the training of robust classifiers: (i) adversarial perturbations align with dominant singular vectors, (ii) adversarial training and data-dependent spectral norm regularization dampen the singular values, and (iii) both training methods give rise to models that are significantly more linear around data than normally trained ones.

## Broader Impact

The existence of adversarial examples, i.e. small perturbations of the input signal, often imperceptible to humans, that are sufficient to induce large changes in the model output, poses a real danger when deep neural networks are deployed in the real world, as potentially safety-critical machine learning systems become vulnerable to attacks that can alter the system's behaviour in malicious ways. Understanding the origin of this vulnerability and / or acquiring an understanding of how to robustify deep neural networks against such attacks thus becomes crucial for a safe and responsible deployment of machine learning systems.

**Who may benefit from this research**

Our work contributes to understanding the origin of this vulnerability in that it sheds new light onto the attack algorithms used to find adversarial examples. It also contributes to building robust machine learning systems in that it allows practitioners to make more informed and well-founded decisions when training robust models.

**Who may be put at a disadvantage from this research**

Our work, like any theoretical work on adversarial examples, may increase the level of understanding of a malevolent person intending to mount adversarial attacks against deployed machine learning systems which may ultimately put the end-users of these systems at risk. We would like to note, however, that the attack algorithms we analyze in our work already exist and that we believe that the knowledge gained from our work is more beneficial to making models more robust than it could possibly be used to designing stronger adversarial attacks.

**Consequences of failure of the system**

Our work does not by itself constitute a system of any kind, other than providing a rigorous mathematical framework within which to better understand adversarial robustness.

## Acknowledgments and Disclosure of Funding

We would like to thank Michael Tschannen, Sebastian Nowozin and Antonio Orvieto for insightful discussions and helpful comments. All authors are directly funded by ETH Zürich.

## Footnotes

[1]Convolutional layers can be constructed as matrix multiplications by converting them into a Toeplitz matrix.

[2]Due to numerical errors, we can safely assume that the dominant singular value is non-degenerate.

[3] $\mathbf{u} = \mathbf{J}_f(\mathbf{x})\mathbf{v}/||\mathbf{J}_f(\mathbf{x})\mathbf{v}||_2$. Compare with Eq. 3 and the (2,2)-operator norm (= spectral norm) in Eq. 8.

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
