[Supplementary Material]

# 7 Appendix

We begin with a short recap on robust optimization in linear regression in **Section 7.1**. In **Section 7.2** we lay out why input gradient regularization and fast gradient method (FGM) based adversarial training cannot in general effectively robustify against iterative adversarial attacks. Similarly, in **Section 7.3** we argue why we do not expect Frobenius norm regularization to work as well as data-dependent spectral norm regularization in robustifying against $\ell_2$-norm bounded iterative adversarial attacks. In **Section 7.4** we analyze the power method like formulation of adversarial training for the softmax cross-entropy loss. The proof of our main Theorem and the corresponding Lemmas can be found in **Sections 7.5 - 7.8**. Additional implementation details can be found in **Sections 7.9 - 7.11**. Additional experimental results are presented from **Section 7.12** on.

## 7.1 Recap: Robust Optimization and Regularization for Linear Regression

In this section, we recapitulate the basic ideas on the relation between robust optimization and regularization presented in [2]. Note that the notation deviates slightly from the main text: most importantly, the perturbations $\triangle$ refer to perturbations of the entire training data $\mathbf{X}$, as is common in robust optimization.

Consider linear regression with additive perturbations $\triangle$ of the data matrix $\mathbf{X}$

$$\min_{\mathbf{w}} \max_{\triangle \in \mathcal{U}} h \left( \mathbf{y} - (\mathbf{X} + \triangle)\mathbf{w} \right), \tag{19}$$

where $h : \mathbb{R}^n \to \mathbb{R}$ denotes a loss function and $\mathcal{U}$ denotes the uncertainty set. A general way to construct $\mathcal{U}$ is as a ball of bounded matrix norm perturbations $\mathcal{U} = \{\triangle : \|\triangle\| \leqslant \lambda\}$. Of particular interest are induced matrix norms

$$\|\mathbf{A}\|_{g,h} := \max_{\mathbf{w}} \left\{ \frac{h(\mathbf{A}\mathbf{w})}{g(\mathbf{w})} \right\}, \tag{20}$$

where $h : \mathbb{R}^n \to \mathbb{R}$ is a semi-norm and $g : \mathbb{R}^d \to \mathbb{R}$ is a norm. It is obvious that if $h$ fulfills the triangle inequality then one can upper bound

$$\begin{aligned}
h \left( \mathbf{y} - (\mathbf{X} + \triangle)\mathbf{w} \right) &\leqslant h(\mathbf{y} - \mathbf{X}\mathbf{w}) + h(\triangle\mathbf{w}) \\
&\leqslant h(\mathbf{y} - \mathbf{X}\mathbf{w}) + \lambda\, g(\mathbf{w}), \quad \forall \triangle \in \mathcal{U},
\end{aligned} \tag{21}$$

by using (a) the triangle inequality and (b) the definition of the matrix norm.

The question then is, under which circumstances both inequalities become equalities at the maximizing $\triangle^*$. It is straightforward to check [2] Theorem 1 that specifically we may choose the rank 1 matrix

$$\triangle^* = \frac{\lambda}{h(\mathbf{r})} \mathbf{r}\mathbf{v}^\top, \tag{22}$$

where

$$\mathbf{r} = \mathbf{y} - \mathbf{X}\mathbf{w}, \quad \mathbf{v} = \arg\max_{\mathbf{v}: g^*(\mathbf{v})=1} \left\{ \mathbf{v}^\top \mathbf{w} \right\}, \tag{23}$$

with $g^*$ as the dual norm. If $h(\mathbf{r}) = 0$ then one can pick any $\mathbf{u}$ for which $h(\mathbf{u}) = 1$ to form $\triangle = \lambda \mathbf{u}\mathbf{v}^\top$ (such a $\mathbf{u}$ has to exist if $h$ is not identically zero). This shows that, for robust linear regression with induced matrix norm uncertainty sets, robust optimization is equivalent to regularization.

## 7.2 On Input Gradient Regularization and Adversarial Robustness

In this Section we will lay out why input gradient regularization and fast-gradient method (FGM) based adversarial training cannot in general effectively robustify against iterative adversarial attacks.

In Section 5.2 "Alignment of adversarial perturbations with singular vectors", we have seen that perturbations of iterative adversarial attacks strongly align with the dominant right singular vectors $\mathbf{v}$ of the Jacobian $\mathbf{J}_f(\mathbf{x})$, see Figure 2 (right). This alignment reflects also what we would expect from theory, since the dominant right singular vector $\mathbf{v}$ precisely defines the direction in input space along which a norm-bounded perturbation induces the maximal amount of signal-gain when propagated through the linearized network, see comment after Equation 8.

Interestingly, this tendency to align with dominant singular vectors explains why input gradient regularization and fast gradient method (FGM) based adversarial training do not sufficiently protect against iterative adversarial attacks, namely because the input gradient, resp. a single power method iteration, do not yield a sufficiently good approximation for the dominant singular vector in general.

In short, data-dependent operator norm regularization and iteartive attacks based adversarial training correspond to multiple forward-backward passes through (the Jacobian of) the network, while input gradient regularization and FGM based adversarial training corresponds to just a single forward-backward pass.

More technically, the right singular vector $\mathbf{v}$ gives the direction in input space that corresponds to the steepest ascent of $f(\mathbf{x})$ *along* the left singular vector $\mathbf{u}$. In input gradient regularization, the logit space direction $\mathbf{u}$ is determined by the loss function (see Section 7.4 for an example using the softmax cross-entropy loss), which in general is however neither equal nor a good enough approximation to the dominant left singular vector $\mathbf{u}$ of $\mathbf{J}_f(\mathbf{x})$.

In other words, if we knew the dominant singular vector $\mathbf{u}$ of $\mathbf{J}_f(\mathbf{x})$, we could compute the direction $\mathbf{v}$ in a single backward-pass. The computation of the dominant singular vector $\mathbf{u}$, however, involves multiple power-method rounds of forward-backward propagation through $\mathbf{J}_f(\mathbf{x})$ in general.

### 7.3 On Frobenius Norm Regularization and Adversarial Robustness

In this Section, we briefly contrast data-dependent spectral norm regularization with Frobenius norm regularization.

As we have seen in Section 4.2, it is the dominant singular vector corresponding to the largest singular value that determines the optimal adversarial perturbation to the Jacobian and hence the maximal amount of signal-gain that can be induced when propagating an $\ell_2$-norm bounded input vector through the linearized network. Writing $\mathbf{J}_f(\mathbf{x}) = \sigma_1 \mathbf{u}_1 \mathbf{v}_1^T + \sigma_2 \mathbf{u}_2 \mathbf{v}_2^T + \ldots$ in SVD form, it is clear that the largest change in output for a given change in input aligns with $\mathbf{v}_1$. This crucial fact is only indirectly captured by regularizing the Frobenius norm, in that the Frobenius norm (= sum of all singular values) is a trivial upper bound on the spectral norm (= largest singular value).

For that reason, we do not expect the Frobenius norm to work as well as data-dependent spectral norm regularization in robustifying against $\ell_2$-norm bounded iterative adversarial attacks.

### 7.4 Adversarial Training with Cross-Entropy Loss

The adversarial loss function determines the logit-space direction $\mathbf{u}_k$ in the power method like formulation of adversarial training in Equation 18.

Let us consider this for the softmax cross-entropy loss, defined as $\ell_{\mathrm{adv}}(y, \mathbf{z}) := -\log(s_y(\mathbf{z}))$,

$$\ell_{\mathrm{adv}}(y, \mathbf{z}) = -\mathbf{z}_y + \log\Big( \sum_{k=1}^{d} \exp(\mathbf{z}_k) \Big) \tag{24}$$

where the softmax is given by

$$s_y(\mathbf{z}) := \frac{\exp(\mathbf{z}_y)}{\sum_{k=1}^{d} \exp(\mathbf{z}_k)} \tag{25}$$

Untargeted $\ell_2$-PGA (forward pass)

$$\Big[ \tilde{\mathbf{u}}_k \leftarrow \nabla_{\mathbf{z}} \ell_{\mathrm{adv}}(y, \mathbf{z})|_{\mathbf{z}=f(\mathbf{x}_{k-1})} \Big]_i = s_i(f(\mathbf{x}_{k-1})) - \delta_{iy} \tag{26}$$

Targeted $\ell_2$-PGA (forward pass)

$$\Big[ \tilde{\mathbf{u}}_k \leftarrow -\nabla_{\mathbf{z}} \ell_{\mathrm{adv}}(y_{\mathrm{adv}}, \mathbf{z})|_{\mathbf{z}=f(\mathbf{x}_{k-1})} \Big]_i = \delta_{iy_{\mathrm{adv}}} - s_i(f(\mathbf{x}_{k-1})) \tag{27}$$

Notice that the logit gradient can be computed in a forward pass by analytically expressing it in terms of the arguments of the loss function.

Interestingly, for a temperature-dependent softmax cross-entropy loss, the logit-space direction becomes a "label-flip" vector in the low-temperature limit (high inverse temperature $\beta \to \infty$) where the softmax $s_y^\beta(\mathbf{z}) := \exp(\beta \mathbf{z}_y)/(\sum_{k=1}^d \exp(\beta \mathbf{z}_k))$ converges to the argmax: $s^\beta(\mathbf{z}) \overset{\beta \to \infty}{\longrightarrow} \arg\max(\mathbf{z})$. E.g. for targeted attacks $\left[\mathbf{u}_k^{\beta \to \infty}\right]_i = \delta_{iy_{\mathrm{adv}}} - \delta_{iy(\mathbf{x}_{k-1})}$. This implies that in the high $\beta$ limit, iterative PGA finds an input space perturbation $\mathbf{v}_k$ that corresponds to the steepest ascent of $f$ along the "label flip" direction $\mathbf{u}_k^{\beta \to \infty}$.

**A note on canonical link functions.** The gradient of the loss w.r.t. the logits of the classifier takes the form "prediction - target" for both the sum-of-squares error as well as the softmax cross-entropy loss. This is in fact a general result of modelling the target variable with a conditional distribution from the exponential family along with a canonical activation function. This means that adversarial attacks try to find perturbations in input space that induce a logit perturbation that aligns with the difference between the current prediction and the attack target.

## 7.5 Gradient of p-Norm

The gradient of any p-norm is given by

$$\nabla_{\mathbf{x}} ||\mathbf{x}||_p = \frac{\mathrm{sign}(\mathbf{x}) \odot |\mathbf{x}|^{p-1}}{||\mathbf{x}||_p^{p-1}} \tag{28}$$

where $\odot$, $\mathrm{sign}(\cdot)$ and $|\cdot|$ denote elementwise product, sign and absolute value.

In this section, we take a closer look at the $p \to \infty$ limit,

$$\lim_{p \to \infty} \nabla_{\mathbf{x}} ||\mathbf{x}||_p = |\mathcal{I}|^{-1} \mathrm{sign}(\mathbf{x}) \odot \mathbf{1}_{\mathcal{I}} \tag{29}$$

with $\mathbf{1}_{\mathcal{I}} = \sum_{i \in \mathcal{I}} \mathbf{e}_i$, where $\mathcal{I} := \{j \in [1,...,n] : |x_j| = ||\mathbf{x}||_\infty\}$ denotes the set of indices at which $\mathbf{x}$ attains its maximum norm and $\mathbf{e}_i$ is the $i$-th canonical unit vector.

The derivation goes as follows. Consider

$$\lim_{p \to \infty} \sum_i \frac{|x_i|^p}{||\mathbf{x}||_\infty^p} = \lim_{p \to \infty} \sum_i \left(\frac{|x_i|}{||\mathbf{x}||_\infty}\right)^p = |\mathcal{I}| \tag{30}$$

Thus

$$\frac{||\mathbf{x}||_p^{p-1}}{||\mathbf{x}||_\infty^{p-1}} = \left(\frac{\sum_i |x_i|^p}{||\mathbf{x}||_\infty^p}\right)^{(p-1)/p} \overset{p \to \infty}{\longrightarrow} |\mathcal{I}| \tag{31}$$

since $|\mathcal{I}|^{(p-1)/p} \overset{p \to \infty}{\longrightarrow} |\mathcal{I}|$. Now consider

$$\frac{\mathrm{sign}(x_i)|x_i|^{p-1}}{||\mathbf{x}||_p^{p-1}} = \frac{\mathrm{sign}(x_i)|x_i|^{p-1}/||\mathbf{x}||_\infty^{p-1}}{||\mathbf{x}||_p^{p-1}/||\mathbf{x}||_\infty^{p-1}} \tag{32}$$

The numerator

$$\mathrm{sign}(x_i)|x_i|^{p-1}/||\mathbf{x}||_\infty^{p-1} \overset{p \to \infty}{\longrightarrow} \begin{cases} 0 & \text{if } i \notin \mathcal{I} \\ \mathrm{sign}(x_i) & \text{if } i \in \mathcal{I} \end{cases} \tag{33}$$

The denominator $||\mathbf{x}||_p^{p-1}/||\mathbf{x}||_\infty^{p-1} \overset{p \to \infty}{\longrightarrow} |\mathcal{I}|$. The rest is clever notation.

## 7.6 Optimal Perturbation to Linear Function

**Lemma 1** (Optimal Perturbation). *Explicit expression for the optimal perturbation to a linear function under an $\ell_p$-norm constraint[4]. Let $\mathbf{z}$ be an arbitrary non-zero vector, e.g. $\mathbf{z} = \nabla_{\mathbf{x}} \ell_{\mathrm{adv}}$, and let $\mathbf{v}$ be a vector of the same dimension. Then,*

$$\mathbf{v}^* = \arg\max_{\mathbf{v}: ||\mathbf{v}||_p \leqslant 1} \mathbf{v}^\top \mathbf{z} = \frac{\mathrm{sign}(\mathbf{z}) \odot |\mathbf{z}|^{p^*-1}}{||\mathbf{z}||_{p*}^{p^*-1}} \tag{34}$$

*where $\odot$, $\mathrm{sign}(\cdot)$ and $|\cdot|$ denote elementwise product, sign and absolute value, and $p^*$ is the Hölder conjugate of $p$, given by $1/p + 1/p^* = 1$. Note that the maximizer $\mathbf{v}^*$ is attained at a $\mathbf{v}$ with $||\mathbf{v}||_p = 1$, since $\mathbf{v}^\top \mathbf{z}$ is linear in $\mathbf{v}$.*

In particular, we have the following special cases, which is also how the Projected Gradient Descent attack is implemented in the cleverhans library [34],

$$\arg\max_{\mathbf{v}: ||\mathbf{v}||_p \leqslant 1} \mathbf{v}^\top \mathbf{z} = \begin{cases} \mathbf{z}/||\mathbf{z}||_2 & \text{for } p = 2 \\ \mathrm{sign}(\mathbf{z}) & \text{for } p = \infty \\ |\mathcal{I}|^{-1}\mathrm{sign}(\mathbf{z}) \odot \mathbf{1}_{\mathcal{I}} & \text{for } p = 1 \end{cases} \tag{35}$$

The optimal $\ell_1$-norm constrained perturbation $\lim_{p* \to \infty} \mathrm{sign}(\mathbf{z}) \odot |\mathbf{z}|^{p^*-1}/||\mathbf{z}||_{p*}^{p^*-1}$ can be taken to be $|\mathcal{I}|^{-1}\mathrm{sign}(\mathbf{z}) \odot \mathbf{1}_{\mathcal{I}}$ with $\mathbf{1}_{\mathcal{I}} = \sum_{i \in \mathcal{I}} \mathbf{e}_i$, where $\mathcal{I} := \{j \in [1,...,n] : |z_j| = ||\mathbf{z}||_\infty\}$ denotes the set of indices at which $\mathbf{z}$ attains its maximum norm, and $\mathbf{e}_i$ is the $i$-th canonical unit-vector. Note that any other convex combination of unit-vectors from the set of indices at which $\mathbf{z}$ attains its maximum absolute value is also a valid optimal $\ell_1$-norm constrained perturbation.

Finally, before we continue with the proof, we would like to note that the above expression for the maximizer has already been stated in [31], although it hasn't been derived there.

*Proof.* By Hölder's inequality, we have for non-zero $\mathbf{z}$,

$$\mathbf{v}^\top \mathbf{z} \leqslant ||\mathbf{v}||_p ||\mathbf{z}||_{p*} \;, \; \text{ w. equality iff } |\mathbf{v}|^p = \gamma |\mathbf{z}|^{p^*} \tag{36}$$

i.e. equality[5] holds if and only if $|\mathbf{v}|^p$ and $|\mathbf{z}|^{p^*}$ are linearly dependent, where $|\cdot|$ denotes elementwise absolute-value, and where $p^*$ is given by $1/p + 1/p^* = 1$. The proportionality constant $\gamma$ is determined by the normalization requirement $||\mathbf{v}||_p = 1$. For $1 < p < \infty$, we have

$$||\mathbf{v}||_p = (\gamma \sum_{i=1}^{n} |z_i|^{p^*})^{1/p} \overset{!}{=} 1 \implies \gamma = ||\mathbf{z}||_{p*}^{-p^*} \tag{37}$$

Thus, equality holds iff $|\mathbf{v}| = |\mathbf{z}|^{p^*/p}/||\mathbf{z}||_{p*}^{p^*/p}$, which implies that $\mathbf{v}^* = \mathrm{sign}(\mathbf{z}) \odot |\mathbf{z}|^{p^*-1}/||\mathbf{z}||_{p*}^{p^*-1}$, since $\mathbf{v}$ must have the same sign as $\mathbf{z}$ and $p^*/p = p^* - 1$. For $p = 1$, $\gamma = (|\mathcal{I}| \, ||\mathbf{z}||_\infty)^{-1}$, where $\mathcal{I} := \{j \in [1,...,n] : |z_j| = ||\mathbf{z}||_\infty\}$, i.e. $|\mathcal{I}|$ counts the multiplicity of the maximum element in the maximum norm. It is easy to see that $|\mathcal{I}|^{-1}\mathrm{sign}(\mathbf{z}) \odot \mathbf{1}_{\mathcal{I}}$ is a maximizer in this case. For $p = \infty$, it is trivially clear that the maximizer $\mathbf{v}^*$ is given by $\mathrm{sign}(\mathbf{z})$. $\square$

As an illustrative sanity-check, let us also verify that the above explicit expression for the optimal perturbation $\mathbf{v}^*$ has $\ell_p$ norm equal to one. Let $\mathbf{z}$ be an arbitrary non-zero vector of the same dimension as $\mathbf{v}$ and let $1 < p < \infty$, then

$$||\mathbf{v}^*||_p = \Big( \sum_{i=1}^{n} \Big| \frac{\text{sign}(z_i) \, |z_i|^{p^*-1}}{||\mathbf{z}||_{p^*}^{p^*-1}} \Big|^p \Big)^{1/p} \tag{38}$$

$$= \Big( \sum_{i=1}^{n} \Big( \frac{|z_i|^{p^*-1}}{||\mathbf{z}||_{p^*}^{p^*-1}} \Big)^p \Big)^{1/p} \tag{39}$$

$$= \Big( \sum_{i=1}^{n} \frac{|z_i|^{p^*}}{||\mathbf{z}||_{p^*}^{p^*}} \Big)^{1/p} = 1 \tag{40}$$

where we have used that $(p^* - 1)p = p^*$. For $p = 1$, we have

$$||\mathbf{v}^*||_1 = \sum_{i=1}^{n} \Big| \frac{1}{|\mathcal{I}|} \text{sign}(z_i) \mathbb{1}_{\{i \in \mathcal{I}\}} \Big| \tag{41}$$

$$= \sum_{i=1}^{n} \Big| \frac{1}{|\mathcal{I}|} \mathbb{1}_{\{i \in \mathcal{I}\}} \Big| = 1 \tag{42}$$

where $\mathbb{1}_{\{\cdot\}}$ denotes the indicator function. For $p = \infty$, we have

$$||\mathbf{v}^*||_\infty = \max_i |\text{sign}(z_i)| = 1 \tag{43}$$

## 7.7 Projection Lemma

In this section, we prove the following intuitive Lemma.

**Lemma 2** (Projection Lemma). *(First part) Let $\mathbf{v}$ and $\tilde{\mathbf{v}} \neq \mathbf{0}$ be two arbitrary (non-zero) vectors of the same dimension. Then*

$$\lim_{\alpha \to \infty} \Pi_{\{||\cdot||_p=1\}}(\mathbf{v} + \alpha \tilde{\mathbf{v}}) = \frac{\text{sign}(\tilde{\mathbf{v}}) \odot |\tilde{\mathbf{v}}|^{p^*-1}}{||\tilde{\mathbf{v}}||_{p^*}^{p^*-1}} \tag{44}$$

*where $\odot$, $\text{sign}(\cdot)$ and $|\cdot|$ denote elementwise product, sign and absolute-value, and where $p^*$ denotes the Hölder conjugate of $p$ defined by $1/p + 1/p^* = 1$. Moreover, if $\tilde{\mathbf{v}}$ is of the form $\tilde{\mathbf{v}} = \text{sign}(\mathbf{z}) \odot |\mathbf{z}|^{p^*-1}/||\mathbf{z}||_{p^*}^{p^*-1}$, for an arbitrary non-zero vector $\mathbf{z}$ and $p \in \{1, 2, \infty\}$, then $\lim_{\alpha \to \infty} \Pi_{\{||\cdot||_p=1\}}(\mathbf{v} + \alpha \tilde{\mathbf{v}}) = \tilde{\mathbf{v}}$. (Second part) Let $\mathbf{x}_{k-1} \in \mathcal{B}_\epsilon^p(\mathbf{x})$ and let $\mathbf{v}_k \neq \mathbf{0}$ be an arbitrary non-zero vector of the same dimension. Then*

$$\mathbf{x}_k = \lim_{\alpha \to \infty} \Pi_{\mathcal{B}_\epsilon^p(\mathbf{x})}(\mathbf{x}_{k-1} + \alpha \mathbf{v}_k) \tag{45}$$

$$= \mathbf{x} + \lim_{\alpha \to \infty} \Pi_{\{||\cdot||_p=\epsilon\}}(\alpha \mathbf{v}_k) \tag{46}$$

$$= \mathbf{x} + \epsilon \frac{\text{sign}(\mathbf{v}_k) \odot |\mathbf{v}_k|^{p^*-1}}{||\mathbf{v}_k||_{p^*}^{p^*-1}} \tag{47}$$

*Moreover, if $\mathbf{v}_k$ is of the form $\mathbf{v}_k = \text{sign}(\mathbf{z}) \odot |\mathbf{z}|^{p^*-1}/||\mathbf{z}||_{p^*}^{p^*-1}$, for $p \in \{1, 2, \infty\}$ and an arbitrary non-zero vector $\mathbf{z}$ (as is the case if $\mathbf{v}_k$ is given by the backward pass in Equation 18), then $\mathbf{x}_k = \mathbf{x} + \epsilon \mathbf{v}_k$.*

*Proof.* **First part.**

$$\lim_{\alpha \to \infty} \Pi_{\{||\cdot||_p=1\}}(\mathbf{v} + \alpha \tilde{\mathbf{v}}) \tag{48}$$

$$= \lim_{\alpha \to \infty} \underset{\mathbf{v}^*:||\mathbf{v}^*||_p=1}{\arg\min} ||\mathbf{v}^* - \mathbf{v} - \alpha \tilde{\mathbf{v}}||_2 \tag{49}$$

$$= \lim_{\alpha \to \infty} \underset{\mathbf{v}^*:||\mathbf{v}^*||_p=1}{\arg\min} (\mathbf{v}^* - \mathbf{v} - \alpha \tilde{\mathbf{v}})^\top (\mathbf{v}^* - \mathbf{v} - \alpha \tilde{\mathbf{v}}) \tag{50}$$

$$= \lim_{\alpha \to \infty} \underset{\mathbf{v}^*:||\mathbf{v}^*||_p=1}{\arg\min} \mathbf{v}^{*\top}\mathbf{v}^* - 2\mathbf{v}^{*\top}\mathbf{v} - 2\alpha\mathbf{v}^{*\top}\tilde{\mathbf{v}} + \text{const} \tag{51}$$

where the const term is independent of $\mathbf{v}^*$ and thus irrelevant for the $\arg\min$. Next, we observe that in the limit $\alpha \to \infty$, the $\mathbf{v}^{*\top}\mathbf{v}^* - 2\mathbf{v}^{*\top}\mathbf{v}$ term vanishes relative to the $\alpha\mathbf{v}^{*\top}\tilde{\mathbf{v}}$ term, hence

$$\lim_{\alpha\to\infty} \Pi_{\{||\cdot||_p=1\}}(\mathbf{v} + \alpha\tilde{\mathbf{v}}) \tag{52}$$

$$= \underset{\mathbf{v}^*:||\mathbf{v}^*||_p=1}{\arg\min} \quad -\mathbf{v}^{*\top}\tilde{\mathbf{v}} \tag{53}$$

$$= \underset{\mathbf{v}^*:||\mathbf{v}^*||_p=1}{\arg\max} \quad \mathbf{v}^{*\top}\tilde{\mathbf{v}} \tag{54}$$

$$= \frac{\text{sign}(\tilde{\mathbf{v}}) \odot |\tilde{\mathbf{v}}|^{p^*-1}}{||\tilde{\mathbf{v}}||_{p^*}^{p^*-1}} \tag{55}$$

where in the last line we have used Equation 16 for the optimal perturbation to a linear function under an $\ell_p$-norm constraint that we have proven in Lemma 1.

Moreover, if $\tilde{\mathbf{v}}$ is of the form $\tilde{\mathbf{v}} = \text{sign}(\mathbf{z}) \odot |\mathbf{z}|^{p^*-1}/||\mathbf{z}||_{p^*}^{p^*-1}$, then

$$\frac{\text{sign}(\tilde{\mathbf{v}}) \odot |\tilde{\mathbf{v}}|^{p^*-1}}{||\tilde{\mathbf{v}}||_{p^*}^{p^*-1}} = \frac{\text{sign}(\mathbf{z}) \odot |\mathbf{z}|^{(p^*-1)(p^*-1)}}{|| \, |\mathbf{z}|^{p^*-1} \, ||_{p^*}^{p^*-1}} \tag{56}$$

Now, observe that for $p \in \{1, 2, \infty\}$, the Hölder conjugate $p^* \in \{1, 2, \infty\}$. In particular, for these values $p^*$ satisfies $(p^*-1)(p^*-1) = p^*-1$ (since $0 \cdot 0 = 0, 1 \cdot 1 = 1, \infty \cdot \infty = \infty$).

Thus, the numerator (and hence the direction) remains the same. Moreover, we also have that $|| \, |\mathbf{z}|^{p^*-1} \, ||_{p^*}^{p^*-1} = ||\mathbf{z}||_{p^*}^{p^*-1}$ (and hence the magnitude remains the same, too). For $p^* = 1$, $|| \, |\mathbf{z}|^0 \, ||_1^0 = ||\mathbf{1}||_1^0 = 1 = ||\mathbf{z}||_1^0$ for any non-zero $\mathbf{z}$. For $p^* = 2$, $|| \, |\mathbf{z}|^1 \, ||_2^1 = ||\mathbf{z}||_2$ for any $\mathbf{z}$. For the $p^* = \infty$ case, we consider the full expression $\text{sign}(\mathbf{z}) \odot |\mathbf{z}|^{(p^*-1)(p^*-1)}/|| \, |\mathbf{z}|^{p^*-1} \, ||_{p^*}^{p^*-1} = |\mathcal{I}'|^{-1}\text{sign}(\mathbf{z}) \odot \mathbf{1}_{\mathcal{I}'}$ where $\mathcal{I}' := \{j \in [1, ..., n] : |z_j|^{p^*-1} = || \, |\mathbf{z}|^{p^*-1} \, ||_\infty\}$ denotes the set of indices at which $|\mathbf{z}|^{p^*-1}$ attains its maximum norm. It is easy to see that $\mathcal{I}' = \mathcal{I}$, i.e. the set of indices of maximal elements remains the same.

Thus, if $\tilde{\mathbf{v}}$ is of the form $\tilde{\mathbf{v}} = \text{sign}(\mathbf{z}) \odot |\mathbf{z}|^{p^*-1}/||\mathbf{z}||_{p^*}^{p^*-1}$, then it is a fix point of Equation 16 and hence of the projection

$$\lim_{\alpha\to\infty} \Pi_{\{||\cdot||_p=1\}}(\mathbf{v} + \alpha\tilde{\mathbf{v}}) = \frac{\text{sign}(\tilde{\mathbf{v}}) \odot |\tilde{\mathbf{v}}|^{p^*-1}}{||\tilde{\mathbf{v}}||_{p^*}^{p^*-1}} = \tilde{\mathbf{v}} \quad \text{for} \ \ p^* \in \{1, 2, \infty\} \tag{57}$$

Another way to see this is by checking that the operation $\text{sign}(\cdot)\odot|\cdot|^{p^*-1}/||\cdot||_{p^*}^{p^*-1}$, for $p^* \in \{1, 2, \infty\}$, leaves the corresponding expressions for the optimal perturbation in the RHS of Equation 35 invariant.

Finally, note how the Projection Lemma implies that

$$\lim_{\alpha\to\infty} \Pi_{\{||\cdot||_p=1\}}(\alpha\mathbf{z}) = \underset{\mathbf{v}:||\mathbf{v}||_p\leqslant 1}{\arg\max} \mathbf{v}^\top\mathbf{z} \,, \tag{58}$$

for an arbitrary non-zero $\mathbf{z}$, which is an interesting result in its own right.

**Second part.**

By definition of the orthogonal projection, we have

$$\mathbf{x}_k = \lim_{\alpha\to\infty} \Pi_{\mathcal{B}_\epsilon^p(\mathbf{x})}(\mathbf{x}_{k-1} + \alpha\mathbf{v}_k) \tag{59}$$

$$= \lim_{\alpha\to\infty} \argmin_{\mathbf{x}^*:||\mathbf{x}^*-\mathbf{x}||_p\leqslant\epsilon} ||\mathbf{x}^*-\mathbf{x}_{k-1}-\alpha\mathbf{v}_k||_2 \tag{60}$$

$$= \lim_{\alpha\to\infty} \argmin_{\mathbf{x}+\mathbf{v}^*:||\mathbf{v}^*||_p\leqslant\epsilon} ||\mathbf{x}+\mathbf{v}^*-\mathbf{x}_{k-1}-\alpha\mathbf{v}_k||_2 \tag{61}$$

$$= \mathbf{x} + \lim_{\alpha\to\infty} \argmin_{\mathbf{v}^*:||\mathbf{v}^*||_p\leqslant\epsilon} ||\mathbf{x}+\mathbf{v}^*-\mathbf{x}_{k-1}-\alpha\mathbf{v}_k||_2^2 \tag{62}$$

$$= \mathbf{x} + \lim_{\alpha\to\infty} \argmin_{\mathbf{v}^*:||\mathbf{v}^*||_p\leqslant\epsilon} \left\{||\mathbf{v}^*-\alpha\mathbf{v}_k||_2^2 + 2\mathbf{v}^{*\top}\mathrm{const}+\mathrm{const}^2\right\} \tag{63}$$

$$= \mathbf{x} + \lim_{\alpha\to\infty} \argmin_{\mathbf{v}^*:||\mathbf{v}^*||_p\leqslant\epsilon} ||\mathbf{v}^* - \alpha\mathbf{v}_k||_2^2 \tag{64}$$

$$= \mathbf{x} + \lim_{\alpha\to\infty} \argmin_{\mathbf{v}^*:||\mathbf{v}^*||_p\leqslant\epsilon} ||\mathbf{v}^* - \alpha\mathbf{v}_k||_2 \tag{65}$$

$$= \mathbf{x} + \lim_{\alpha\to\infty} \argmin_{\mathbf{v}^*:||\mathbf{v}^*||_p=\epsilon} ||\mathbf{v}^* - \alpha\mathbf{v}_k||_2 \tag{66}$$

$$= \mathbf{x} + \lim_{\alpha\to\infty} \Pi_{\{||\cdot||_p=\epsilon\}}(\alpha\mathbf{v}_k) \tag{67}$$

where (i) we have used that the $\mathrm{const}^2$ term is independent of $\mathbf{v}^*$ and thus irrelevant for the $\argmin$, (ii) in the fourth-to-last line we have dropped all the terms that vanish relative to the limit $\alpha \to \infty$, and (iii) since $\alpha\mathbf{v}_k$ is outside the $\ell_p$-ball in the limit $\alpha \to \infty$, projecting into the $\ell_p$-ball $\{\mathbf{v}^* : ||\mathbf{v}^*||_p \leqslant 1\}$ is equivalent to projecting onto its boundary $\{\mathbf{v}^* : ||\mathbf{v}^*||_p = 1\}$.

By the first part of the Projection Lemma, we also have that

$$\mathbf{x}_k = \mathbf{x} + \lim_{\alpha\to\infty} \Pi_{\{||\cdot||_p=\epsilon\}}(\alpha\mathbf{v}_k) \tag{68}$$

$$= \mathbf{x} + \epsilon\, \frac{\mathrm{sign}(\mathbf{v}_k) \odot |\mathbf{v}_k|^{p^*-1}}{||\mathbf{v}_k||_{p^*}^{p^*-1}} \tag{69}$$

$$= \mathbf{x} + \epsilon\mathbf{v}_k \quad \text{if} \quad \mathbf{v}_k = \frac{\mathrm{sign}(\mathbf{z}) \odot |\mathbf{z}|^{p^*-1}}{||\mathbf{z}||_{p^*}^{p^*-1}} \tag{70}$$

for an arbitrary non-zero vector $\mathbf{z}$. This completes the proof of the Lemma.

For illustrative purposes, we provide an alternative proof of the second part of the Lemma for $p \in \{2, \infty\}$, because for these values the projection can be written down explicitly away from the $\alpha \to \infty$ limit (the $p = 1$ projection can be written down in a simple form in the $\alpha \to \infty$ limit only).

We first consider the **case** $p = 2$. The $\ell_2$-norm ball projection can be expressed as follows,

$$\Pi_{\mathcal{B}_\epsilon^2(\mathbf{x})}(\tilde{\mathbf{x}}_k) = \mathbf{x} + \epsilon(\tilde{\mathbf{x}}_k - \mathbf{x})/\max(\epsilon, ||\tilde{\mathbf{x}}_k - \mathbf{x}||_2), \tag{71}$$

where $\tilde{\mathbf{x}}_k = \mathbf{x}_{k-1} + \alpha\mathbf{v}_k$ and where the $\max(\epsilon, ||\tilde{\mathbf{x}}_k - \mathbf{x}||_2)$ ensures that if $||\tilde{\mathbf{x}}_k - \mathbf{x}||_2 < \epsilon$ then $\mathbf{x}_k = \tilde{\mathbf{x}}_k$, i.e. we only need to project $\tilde{\mathbf{x}}_k$ if it is outside the $\epsilon$-ball $\mathcal{B}_\epsilon^2(\mathbf{x})$.

Thus, in the limit $\alpha \to \infty$,

$$\lim_{\alpha \to \infty} \Pi_{\mathcal{B}_\epsilon^2(\mathbf{x})}(\mathbf{x}_{k-1} + \alpha \mathbf{v}_k) \tag{72}$$

$$= \lim_{\alpha \to \infty} \mathbf{x} + \epsilon \frac{\tilde{\mathbf{x}}_k - \mathbf{x}}{\max(\epsilon, ||\tilde{\mathbf{x}}_k - \mathbf{x}||_2)} \tag{73}$$

$$= \mathbf{x} + \epsilon \lim_{\alpha \to \infty} \frac{\mathbf{x}_{k-1} + \alpha \mathbf{v}_k - \mathbf{x}}{\max(\epsilon, ||\mathbf{x}_{k-1} + \alpha \mathbf{v}_k - \mathbf{x}||_2)} \tag{74}$$

$$= \mathbf{x} + \epsilon \lim_{\alpha \to \infty} \frac{\alpha(\mathbf{v}_k + \frac{1}{\alpha}(\mathbf{x}_{k-1} - \mathbf{x}))}{\max(\epsilon, \alpha||\mathbf{v}_k + \frac{1}{\alpha}(\mathbf{x}_{k-1} - \mathbf{x})||_2)} \tag{75}$$

$$= \mathbf{x} + \epsilon \lim_{\alpha \to \infty} \frac{\alpha(\mathbf{v}_k + \frac{1}{\alpha}(\mathbf{x}_{k-1} - \mathbf{x}))}{\alpha||\mathbf{v}_k + \frac{1}{\alpha}(\mathbf{x}_{k-1} - \mathbf{x})||_2} \tag{76}$$

$$= \mathbf{x} + \epsilon \lim_{\alpha \to \infty} \Pi_{\{||\cdot||_2 = 1\}}(\alpha \mathbf{v}_k) \tag{77}$$

$$= \mathbf{x} + \epsilon \mathbf{v}_k / ||\mathbf{v}_k||_2 \tag{78}$$

$$= \mathbf{x} + \epsilon \mathbf{v}_k \quad \text{if } \mathbf{v}_k = \tilde{\mathbf{v}}_k / ||\tilde{\mathbf{v}}_k||_2 \tag{79}$$

where in the fourth-to-last line we used that the max will be attained at its second argument in the limit $\alpha \to \infty$ since $||\mathbf{v}_k + \frac{1}{\alpha}(\mathbf{x}_{k-1} - \mathbf{x})||_2 > 0$ for $\mathbf{v}_k \neq 0$, and the last line holds if $\mathbf{v}_k$ is of the form $\tilde{\mathbf{v}}_k / ||\tilde{\mathbf{v}}_k||_2$.

Next, we consider the **case** $p = \infty$. The $\ell_\infty$-norm ball projection (clipping) can be expressed as follows,

$$\Pi_{\mathcal{B}_\epsilon^\infty(\mathbf{x})}(\tilde{\mathbf{x}}_k) = \mathbf{x} + \max(-\epsilon, \min(\epsilon, \tilde{\mathbf{x}}_k - \mathbf{x})), \tag{80}$$

where $\tilde{\mathbf{x}}_k = \mathbf{x}_{k-1} + \alpha \mathbf{v}_k$ and where the max and min are taken *elementwise*. Note that the order of the max and min operators can be exchanged, as we prove in the "min-max-commutativity" Lemma 3 below.

Thus, in the limit $\alpha \to \infty$,

$$\lim_{\alpha \to \infty} \Pi_{\mathcal{B}_\epsilon^\infty(\mathbf{x})}(\mathbf{x}_{k-1} + \alpha \mathbf{v}_k) - \mathbf{x} \tag{81}$$

$$= \lim_{\alpha \to \infty} \max(-\epsilon, \min(\epsilon, \alpha \mathbf{v}_k + \mathbf{x}_{k-1} - \mathbf{x})) \tag{82}$$

$$= \lim_{\alpha \to \infty} \Big\{ \mathbb{1}_{\{\text{sign}(\mathbf{v}_k) > 0\}} \max(-\epsilon, \min(\epsilon, \alpha|\mathbf{v}_k| + \mathbf{x}_{k-1} - \mathbf{x}))$$
$$+ \mathbb{1}_{\{\text{sign}(\mathbf{v}_k) < 0\}} \max(-\epsilon, \min(\epsilon, -\alpha|\mathbf{v}_k| + \mathbf{x}_{k-1} - \mathbf{x})) \Big\} \tag{83}$$

where in going from the second to the third line we used that $\mathbf{v}_k = \text{sign}(\mathbf{v}_k) \odot |\mathbf{v}_k|$.

Next, observe that

$$\lim_{\alpha \to \infty} \max(-\epsilon, \min(\epsilon, \alpha|\mathbf{v}_k| + \mathbf{x}_{k-1} - \mathbf{x})) \tag{84}$$

$$= \lim_{\alpha \to \infty} \min(\epsilon, \alpha|\mathbf{v}_k| + \mathbf{x}_{k-1} - \mathbf{x}) = \epsilon \tag{85}$$

since $\min(\epsilon, \alpha|\mathbf{v}_k| + \mathbf{x}_{k-1} - \mathbf{x}) > -\epsilon$.

Similarly, we have

$$\lim_{\alpha \to \infty} \max(-\epsilon, \min(\epsilon, -\alpha|\mathbf{v}_k| + \mathbf{x}_{k-1} - \mathbf{x})) \tag{86}$$

$$= \lim_{\alpha \to \infty} \min(\epsilon, \max(-\epsilon, -\alpha|\mathbf{v}_k| + \mathbf{x}_{k-1} - \mathbf{x})) \tag{87}$$

$$= \lim_{\alpha \to \infty} \max(-\epsilon, -\alpha|\mathbf{v}_k| + \mathbf{x}_{k-1} - \mathbf{x}) \tag{88}$$

$$= -\epsilon \tag{89}$$

where for the first equality we have used the "min-max-commutativity" Lemma 3 below, which asserts that the order of the max and min can be exchanged, while the second equality holds since $\max(-\epsilon, -\alpha|\mathbf{v}_k| + \mathbf{x}_{k-1} - \mathbf{x}) < \epsilon$.

With that, we can continue

$$\lim_{\alpha \to \infty} \Pi_{\mathcal{B}_\epsilon^\infty(\mathbf{x})}(\mathbf{x}_{k-1} + \alpha \mathbf{v}_k) \tag{90}$$

$$= \mathbf{x} + \epsilon \mathbb{1}_{\{\text{sign}(\mathbf{v}_k) > 0\}} - \epsilon \mathbb{1}_{\{\text{sign}(\mathbf{v}_k) < 0\}} \tag{91}$$

$$= \mathbf{x} + \epsilon \, \text{sign}(\mathbf{v}_k) \tag{92}$$

$$= \mathbf{x} + \epsilon \lim_{\alpha \to \infty} \Pi_{\{||\cdot||_\infty = 1\}}(\alpha \mathbf{v}_k) \tag{93}$$

$$= \mathbf{x} + \epsilon \, \mathbf{v}_k \quad \text{if } \mathbf{v}_k = \text{sign}(\tilde{\mathbf{v}}_k) \tag{94}$$

where the last line holds if $\mathbf{v}_k$ is of the form $\text{sign}(\mathbf{z})$, since $\text{sign}(\mathbf{v}_k) = \text{sign}(\text{sign}(\tilde{\mathbf{v}}_k)) = \text{sign}(\tilde{\mathbf{v}}_k) = \mathbf{v}_k$. Note that the last line can directly be obtained from the third-to-last line if $\mathbf{v}_k = \text{sign}(\tilde{\mathbf{v}}_k)$.

Finally, for the **general case**, we provide the following additional intuition. Taking the limit $\alpha \to \infty$ has two effects: firstly, it puts all the weight in the sum $\mathbf{x}_{k-1} + \alpha \mathbf{v}_k$ on $\alpha \mathbf{v}_k$ and secondly, it takes every component of $\mathbf{v}_k$ out of the $\epsilon$-ball $\mathcal{B}_\epsilon^p(\mathbf{x})$. As a result, $\Pi_{\mathcal{B}_\epsilon^p(\mathbf{x})}$ will project $\alpha \mathbf{v}_k$ onto a point on the boundary of the $\epsilon$-ball, which is precisely the set $\{\mathbf{v} : ||\mathbf{v}||_p = \epsilon\}$. Hence, $\lim_{\alpha \to \infty} \Pi_{\mathcal{B}_\epsilon^p(\mathbf{x})}(\mathbf{x}_{k-1} + \alpha \mathbf{v}_k) = \mathbf{x} + \lim_{\alpha \to \infty} \Pi_{\{||\cdot||_p = \epsilon\}}(\alpha \mathbf{v}_k)$. $\qquad \square$

Finally, we provide another very intuitive yet surprisingly hard[6] to prove result:

**Lemma 3** (Min-max-commutativity). *The order of the* elementwise max *and* min *in the projection (clipping) operator* $\Pi_{\mathcal{B}_\epsilon^\infty(\mathbf{x})}$ *can be exchanged, i.e.*

$$\max(-\epsilon, \min(\epsilon, \mathbf{x})) = \min(\epsilon, \max(-\epsilon, \mathbf{x})) \tag{95}$$

*Proof.* We are using the following representations which hold *elementwise* for all $\mathbf{a}, \mathbf{x} \in \mathbb{R}^n$:

$$\max(\mathbf{a}, \mathbf{x}) = \mathbf{a} + \max(0, \mathbf{x} - \mathbf{a}) = \mathbf{a} + \mathbb{1}_{\{\mathbf{a} < \mathbf{x}\}}(\mathbf{x} - \mathbf{a}) \tag{96}$$

$$\min(\mathbf{a}, \mathbf{x}) = \mathbf{a} + \min(0, \mathbf{x} - \mathbf{a}) = \mathbf{a} + \mathbb{1}_{\{\mathbf{x} < \mathbf{a}\}}(\mathbf{x} - \mathbf{a}) \tag{97}$$

where $\mathbb{1}_{\{\cdot\}}$ denotes the elementwise indicator function.

With these, we have

$$\max(-\epsilon, \min(\epsilon, \mathbf{x})) \tag{98}$$

$$= -\epsilon + \mathbb{1}_{\{-\epsilon < \min(\epsilon, \mathbf{x})\}}(\min(\epsilon, \mathbf{x}) + \epsilon) \tag{99}$$

$$= -\epsilon + \mathbb{1}_{\{-\epsilon < \mathbf{x}\}}(2\epsilon + \mathbb{1}_{\{\mathbf{x} < \epsilon\}}(\mathbf{x} - \epsilon)) \tag{100}$$

$$= -\epsilon + 2\epsilon \mathbb{1}_{\{-\epsilon < \mathbf{x}\}} + \mathbb{1}_{\{-\epsilon < \mathbf{x} < \epsilon\}}(\mathbf{x} - \epsilon) \tag{101}$$

$$= -\epsilon + 2\epsilon(\mathbb{1}_{\{-\epsilon < \mathbf{x} < \epsilon\}} + \mathbb{1}_{\{\epsilon < \mathbf{x}\}}) + \mathbb{1}_{\{-\epsilon < \mathbf{x} < \epsilon\}}(\mathbf{x} - \epsilon) \tag{102}$$

$$= -\epsilon + 2\epsilon \mathbb{1}_{\{\epsilon < \mathbf{x}\}} + \mathbb{1}_{\{-\epsilon < \mathbf{x} < \epsilon\}}(\mathbf{x} + \epsilon) \tag{103}$$

$$= -\epsilon + 2\epsilon(1 - \mathbb{1}_{\{\mathbf{x} < \epsilon\}}) + \mathbb{1}_{\{-\epsilon < \mathbf{x} < \epsilon\}}(\mathbf{x} + \epsilon) \tag{104}$$

$$= \epsilon + \mathbb{1}_{\{\mathbf{x} < \epsilon\}}(-2\epsilon + \mathbb{1}_{\{-\epsilon < \mathbf{x}\}}(\mathbf{x} + \epsilon)) \tag{105}$$

$$= \epsilon + \mathbb{1}_{\{\max(-\epsilon, \mathbf{x}) < \epsilon\}}(\max(-\epsilon, \mathbf{x}) - \epsilon) \tag{106}$$

$$= \min(\epsilon, \max(-\epsilon, \mathbf{x})) \tag{107}$$

where we used that $\mathbb{1}_{\{-\epsilon < \min(\epsilon, \mathbf{x})\}} = \mathbb{1}_{\{-\epsilon < \mathbf{x}\}}$ and $\mathbb{1}_{\{\max(-\epsilon, \mathbf{x}) < \epsilon\}} = \mathbb{1}_{\{\mathbf{x} < \epsilon\}}$. $\qquad \square$

## 7.8 Proof of Main Theorem

The conditions on $\epsilon$ and $\alpha$ can be considered specifics of the respective iteration method. The condition that $\epsilon$ be small enough such that $\mathcal{B}^p_\epsilon(\mathbf{x})$ is contained in the ReLU cell around $\mathbf{x}$ ensures that $\mathbf{J}_f(\mathbf{x}^*) = \mathbf{J}_f(\mathbf{x})$ for all $\mathbf{x}^* \in \mathcal{B}^p_\epsilon(\mathbf{x})$. The power-method limit $\alpha \to \infty$ means that in the update equations all the weight (no weight) is placed on the current gradient direction (previous iterates).

To proof the theorem we need to show that the updates for $\ell_p$-norm constrained projected gradient ascent based adversarial training with an $\ell_q$-norm loss on the logits reduce to the corresponding updates for data-dependent operator norm regularization in Equation 12 under the above conditions on $\epsilon$ and $\alpha$.

*Proof.* For an $\ell_q$-norm loss on the logits of the clean and perturbed input $\ell_{\mathrm{adv}}(f(\mathbf{x}), f(\mathbf{x}^*)) = ||f(\mathbf{x}) - f(\mathbf{x}^*)||_q$, the corresponding $\ell_p$-norm constrained projected gradient ascent updates in Equation 18 are

$$
\begin{aligned}
\mathbf{u}_k &\leftarrow \frac{\mathrm{sign}(\tilde{\mathbf{u}}_k) \odot |\tilde{\mathbf{u}}_k|^{q-1}}{||\tilde{\mathbf{u}}_k||_q^{q-1}} \;,\quad \tilde{\mathbf{u}}_k \leftarrow f(\mathbf{x}_{k-1}) - f(\mathbf{x}) \\
\mathbf{v}_k &\leftarrow \frac{\mathrm{sign}(\tilde{\mathbf{v}}_k) \odot |\tilde{\mathbf{v}}_k|^{p^*-1}}{||\tilde{\mathbf{v}}_k||_{p*}^{p^*-1}} \;,\quad \tilde{\mathbf{v}}_k \leftarrow \mathbf{J}_f(\mathbf{x}_{k-1})^\top \mathbf{u}_k \\
\mathbf{x}_k &\leftarrow \Pi_{\mathcal{B}^p_\epsilon(\mathbf{x})}(\mathbf{x}_{k-1} + \alpha \mathbf{v}_k)
\end{aligned}
\tag{108}
$$

In the limit $\alpha \to \infty$, $\mathbf{x}_{k-1} = \mathbf{x} + \epsilon \mathbf{v}_{k-1}$ by the "Projection Lemma" (Lemma 2) and thus for small enough $\epsilon$, $f(\mathbf{x}_{k-1}) - f(\mathbf{x}) = \mathbf{J}_f(\mathbf{x})(\mathbf{x}_{k-1} - \mathbf{x}) = \epsilon \mathbf{J}_f(\mathbf{x}) \mathbf{v}_{k-1}$ (equality holds because $\mathbf{x}_{k-1} \in \mathcal{B}^p_\epsilon(\mathbf{x}) \subset X(\phi_\mathbf{x})$). Thus, the forward pass becomes

$$
\mathbf{u}_k \leftarrow \frac{\mathrm{sign}(\tilde{\mathbf{u}}_k) \odot |\tilde{\mathbf{u}}_k|^{q-1}}{||\tilde{\mathbf{u}}_k||_q^{q-1}} \;,\quad \tilde{\mathbf{u}}_k \leftarrow \mathbf{J}_f(\mathbf{x}) \mathbf{v}_{k-1}
\tag{109}
$$

For the backward pass, we have

$$
\tilde{\mathbf{v}}_k = \mathbf{J}_f(\mathbf{x}_{k-1})^\top \mathbf{u}_k = \mathbf{J}_f(\mathbf{x})^\top \mathbf{u}_k \,,
\tag{110}
$$

since $\mathbf{J}_f(\mathbf{x}_k) = \mathbf{J}_f(\mathbf{x})$ for all $\mathbf{x}_k \in \mathcal{B}^p_\epsilon(\mathbf{x}) \subset X(\phi_\mathbf{x})$. Note that the update equation for $\mathbf{x}_k$ is not needed since the Jacobians in the forward and backward passes don't depend on $\mathbf{x}_k$ for $\mathcal{B}^p_\epsilon(\mathbf{x}) \subset X(\phi_\mathbf{x})$. The update equations for $\ell_p$-norm constrained projected gradient ascent based adversarial training with an $\ell_q$-norm loss on the logits can therefore be written as

$$
\begin{aligned}
\mathbf{u}_k &\leftarrow \mathrm{sign}(\tilde{\mathbf{u}}_k) \odot |\tilde{\mathbf{u}}_k|^{q-1}/||\tilde{\mathbf{u}}_k||_q^{q-1} \;,\quad \tilde{\mathbf{u}}_k \leftarrow \mathbf{J}_f(\mathbf{x}) \mathbf{v}_{k-1} \\
\mathbf{v}_k &\leftarrow \mathrm{sign}(\tilde{\mathbf{v}}_k) \odot |\tilde{\mathbf{v}}_k|^{p^*-1}/||\tilde{\mathbf{v}}_k||_{p*}^{p^*-1},\quad \tilde{\mathbf{v}}_k \leftarrow \mathbf{J}_f(\mathbf{x})^\top \mathbf{u}_k
\end{aligned}
\tag{111}
$$

which is precisely the power method limit of (p, q) operator norm regularization in Equation 12. We have thus shown that the update equations to compute the adversarial perturbation and the data-dependent operator norm maximizer are exactly the same.

It is also easy to see that the objective functions used to update the network parameters for $\ell_p$-norm constrained projected gradient ascent based adversarial training with an $\ell_q$-norm loss on the logits of clean and adversarial inputs in Equation 14

$$
\mathbf{E}_{(\mathbf{x},y)\sim\hat{P}}\left[\ell(y, f(\mathbf{x})) + \lambda \max_{\mathbf{x}^* \in \mathcal{B}^p_\epsilon(\mathbf{x})} ||f(\mathbf{x}) - f(\mathbf{x}^*)||_q\right]
\tag{112}
$$

is by the condition $\mathcal{B}^p_\epsilon(\mathbf{x}) \subset X(\phi_\mathbf{x})$ and $\mathbf{x}^* = \mathbf{x} + \epsilon \mathbf{v}$

$$
\mathbf{E}_{(\mathbf{x},y)\sim\hat{P}}\left[\ell(y, f(\mathbf{x})) + \lambda\epsilon \max_{\mathbf{v}^*:||\mathbf{v}^*||_p \leqslant 1} ||\mathbf{J}_{f(\mathbf{x})}\mathbf{v}||_q\right]
\tag{113}
$$

the same as that of data-dependent (p, q) operator norm regularization in Equation 13.

$\square$

We conclude this section with a note on generalizing our Theorem to allow for activation pattern changes. Proving such an extension for the approximate correspondence between adversarial training and data-dependent operator norm regularization that we observe in our experiments is highly non-trivial, as this requires to take into account how much "nearby" Jacobians can change based on the crossings of ReLU boundaries, which is complicated by the fact that the impact of such crossings depends heavily on the specific activation pattern at input $\mathbf{x}$ and the precise values of the weights and biases in the network. We consider this to be an interesting avenue for future investigations.

## 7.9 Extracting Jacobian as a Matrix

Since we know that any neural network with its nonlinear activation function set to fixed values represents a linear operator, which, locally, is a good approximation to the neural network itself, we develop a method to fully extract and specify this linear operator in the neighborhood of any input datapoint $\mathbf{x}$. We have found the naive way of determining each entry of the linear operator by consecutively computing changes to individual basis vectors to be numerically unstable and therefore have settled for a more robust alternative:

In a first step, we run a set of randomly perturbed versions of $\mathbf{x}$ through the network (with fixed activation functions) and record their outputs at the particular layer that is of interest to us (usually the logit layer). In a second step, we compute a linear regression on these input-output pairs to obtain a weight matrix $\mathbf{W}$ as well as a bias vector $\mathbf{b}$, thereby fully specifying the linear operator. The singular vectors and values of $\mathbf{W}$ can be obtained by performing an SVD.

## 7.10 Dataset, Architecture & Training Methods

We trained Convolutional Neural Networks (CNNs) with seven hidden layers and batch normalization on the CIFAR10 data set [23]. The CIFAR10 dataset consists of 60k $32 \times 32$ colour images in 10 classes, with 6k images per class. It comes in a pre-packaged train-test split, with 50k training images and 10k test images, and can readily be downloaded from `https://www.cs.toronto.edu/~kriz/cifar.html`.

We conduct our experiments on a pre-trained standard convolutional neural network, employing 7 convolutional layers, augmented with BatchNorm, ReLU nonlinearities and MaxPooling. The network achieves 93.5% accuracy on a clean test set. Relevant links to download the pre-trained model can be found in our codebase. For the robustness experiments, we also train a state-of-the-art Wide Residual Net (WRN-28-10) [51]. The network achieves 96.3% accuracy on a clean test set.

We adopt the following standard preprocessing and data augmentation scheme: Each training image is zero-padded with four pixels on each side, randomly cropped to produce a new image with the original dimensions and horizontally flipped with probability one half. We also standardize each image to have zero mean and unit variance when passing it to the classifier.

We train each classifier with a number of different training methods: (i) 'Standard': standard empirical risk minimization with a softmax cross-entropy loss, (ii) 'Adversarial': $\ell_2$-norm constrained projected gradient ascent (PGA) based adversarial training with a softmax cross-entropy loss, (iii) 'global SNR': global spectral norm regularization à la Yoshida & Miyato [50], and (iv) 'd.-d. SNR': data-dependent spectral norm regularization. For the robustness experiments, we also train a state-of-the-art Wide Residual Network (WRN-28-10) [51].

As a default attack strategy we use an $\ell_2$- & $\ell_\infty$-norm constrained PGA white-box attack with cross-entropy adversarial loss $\ell_{\mathrm{adv}}$ and 10 attack iterations. We verified that all our conclusions also hold for larger numbers of attack iterations, however, due to computational constraints we limit the attack iterations to 10. The attack strength $\epsilon$ used for PGA was chosen to be the smallest value such that almost all adversarially perturbed inputs to the standard model are successfully misclassified, which is $\epsilon = 1.75$ (for $\ell_2$-norm) and $\epsilon = 8/255$ (for $\ell_\infty$-norm).

The regularization constants of the other training methods were then chosen in such a way that they roughly achieve the same test set accuracy on clean examples as the adversarially trained model does, i.e. we allow a comparable drop in clean accuracy for regularized and adversarially trained models. When training the derived regularized models, we started from a pre-trained checkpoint and ran a hyper-parameter search over number of epochs, learning rate and regularization constants.

Table 2: CIFAR10 test set accuracies and hyper-parameters for the models and training methods we considered. The regularization constants were chosen such that the models achieve roughly the same accuracy on clean test examples as the adversarially trained model does. See Table 3 for the full hyper-parameter sweep.

| MODEL & TRAINING METHOD | ACC | HYPER-PARAMETERS |
|---|---|---|
| *CNN7* | | |
| STANDARD TRAINING | 93.5% | — |
| ADVERSARIAL TRAINING ($\ell_2 - norm$) | 83.6% | $\epsilon = 1.75, \alpha = 2\epsilon/$ITERS, ITERS $= 10$ |
| ADVERSARIAL TRAINING ($\ell_\infty - norm$) | 82.9% | $\epsilon = 8/255, \alpha = 2\epsilon/$ITERS, ITERS $= 10$ |
| DATA-DEP. SPECTRAL NORM | 84.6% | $\lambda = 3 \cdot 10^{-2}, \epsilon = 1.75$, ITERS $= 10$ |
| DATA-DEP. OPERATOR NORM | 83.0% | $\lambda = 3 \cdot 10^{-2}, \epsilon = 8/255$, ITERS $= 10$ |
| GLOBAL SPECTRAL NORM | 81.5% | $\lambda = 3 \cdot 10^{-4}$, ITERS $= 1, 10$ |
| *WRN-28-10* | | |
| STANDARD TRAINING | 96.3% | — |
| ADVERSARIAL TRAINING ($\ell_2 - norm$) | 91.8% | $\epsilon = 1.75, \alpha = 2\epsilon/$ITERS, ITERS $= 10$ |
| DATA-DEP. SPECTRAL NORM | 91.3% | $\lambda = 3 \cdot 10^{-1}, \epsilon = 1.75$, ITERS $= 10$ |

Table 2 summarizes the test set accuracies and hyper-parameters for all the training methods we considered.

## 7.11 Hyperparameter Sweep

Table 3: Hyperparameter sweep during training. We report results for the best performing models in the main text.

| TRAINING METHOD | HYPERPARAMETER | VALUES TESTED |
|---|---|---|
| ADVERSARIAL TRAINING | $\epsilon$ ($\ell_2$-NORM) | 0.5, 0.75, 1.0, 1.25, 1.5, 1.75, 2.0, 2.5, 2.75, 3.0, 3.25, 3.5, 3.75, 4.0 |
| | $\epsilon$ ($\ell_\infty$-NORM) | 1/255, 2/255, 3/255, 4/255, 5/255, 6/255, 7/255, 8/255, 9/255, 10/255, 11/255, 12/255, 13/255, 14/255, 15/255, 16/255, 17/255, 18/255 |
| | $\alpha$ | $\epsilon$/ITERS, $2\epsilon$/ITERS, $3\epsilon$/ITERS, $4\epsilon$/ITERS, $5\epsilon$/ITERS |
| | ITERS | 1, 2, 3, 5, 8, 10, 15, 20, 30, 40, 50 |
| GLOBAL SPECTRAL NORM REG. | $\lambda$ | $1 \cdot 10^{-5}, 3 \cdot 10^{-5}, 1 \cdot 10^{-4}, 3 \cdot 10^{-4},$ $1 \cdot 10^{-3}, 3 \cdot 10^{-3}, 1 \cdot 10^{-2}, 3 \cdot 10^{-2}, 1 \cdot 10^{-1}, 3 \cdot 10^{-1},$ $1 \cdot 10^{0}, 3 \cdot 10^{0}, 1 \cdot 10^{1}, 3 \cdot 10^{1}$ |
| | ITERS | 1, 10 |
| DATA-DEP. SPECTRAL NORM REG. | $\lambda$ | $1 \cdot 10^{-5}, 3 \cdot 10^{-5}, 1 \cdot 10^{-4}, 3 \cdot 10^{-4},$ $1 \cdot 10^{-3}, 3 \cdot 10^{-3}, 1 \cdot 10^{-2}, 3 \cdot 10^{-2}, 1 \cdot 10^{-1}, 3 \cdot 10^{-1},$ $1 \cdot 10^{0}, 3 \cdot 10^{0}, 1 \cdot 10^{1}, 3 \cdot 10^{1}$ |
| | ITERS | 1, 2, 3, 5, 8, 10, 15, 20, 30, 40, 50 |

# Further Experimental Results

## 7.12 Adversarial Training with Large $\alpha$

Figure 6 shows the result of varying $\alpha$ in adversarial training. As can be seen, the adversarial robustness initially rises with increasing $\alpha$, but after some threshold it levels out and does not change significantly even at very large values.

Figure 6: Test set accuracy on clean and adversarial examples for models adversarially trained with different PGA step sizes $\alpha$. The dashed line indicates the $\alpha$ used when generating adversarial examples at test time. The $\epsilon$ in the AT projection was fixed to the value used in the main text.

## 7.13 Interpolating between AT and d.d. SNR

Figure 7: Test set accuracy on clean and adversarial examples for different networks from scratch each with an objective function that convexly combines adversarial training with data-dependent spectral norm regularization in a way that allows us to interpolate between (i) the fraction of adversarial examples relative to clean examples used during adversarial training controlled by $\lambda$ in Eq. 14 and (ii) the regularization parameter $\tilde{\lambda}$ in Eq. 13. The plot confirms that we can continuously trade-off the contribution of AT with that of d.d. SNR in the empirical risk minimization.

## 7.14 Alignment of Adversarial Perturbations with Dominant Singular Vector

Figure 8 shows the cosine-similarity of adversarial perturbations of mangitude $\epsilon$ with the dominant singular vector of $\mathbf{J}_f(\mathbf{x})$, as a function of perturbation magnitude $\epsilon$. For comparison, we also include the alignment with random perturbations. For all training methods, the larger the perturbation magnitude $\epsilon$, the lesser the adversarial perturbation aligns with the dominant singular vector of $\mathbf{J}_f(\mathbf{x})$, which is to be expected for a simultaneously increasing deviation from linearity. The alignment is similar for adversarially trained and data-dependent spectral norm regularized models and for both larger than that of global spectral norm regularized and naturally trained models.

Figure 8: Alignment of adversarial perturbations with dominant singular vector of $\mathbf{J}_f(\mathbf{x})$ as a function of perturbation magnitude $\epsilon$. The dashed vertical line indicates the $\epsilon$ used during adversarial training. Curves were aggregated over 2000 test samples.

## 7.15 Activation Patterns

Figure 9: Fraction of shared activations as a function of perturbation magnitude $\epsilon$ between activation patterns $\phi_{\mathbf{x}}$ and $\phi_{\mathbf{x}*}$, where $\mathbf{x}$ is a data point sampled from the test set, and x* is an adversarially perturbed input, with perturbation magnitude $\epsilon$.

## 7.16 Global SNR with 10 iterations

In the main section, we have implemented the baseline version of global SNR as close as possible to the descriptions in [50]. However, this included a recommendation from the authors to perform only a single update iteration to the spectral decompositions of the weight matrices per training step. As this is computationally less demanding than the 10 iterations per training step spent on both adversarial training, as well as data-dependent spectral norm regularization, we verify that performing 10 iterations makes no difference to the method of [50]. Figures 10 and 11 reproduce the curves for global SNR from the main part (having used 1 iteration) and overlap it with the same experiments, but done with global SNR using 10 iterations. As can be seen, there is no significant difference.

Figure 10: (Left) Deviation from linearity $||\phi^{L-1}(\mathbf{x} + \mathbf{z}) - (\phi^{L-1}(\mathbf{x}) + \mathbf{J}_{\phi^{L-1}}(\mathbf{x})\mathbf{z})||_2$ as a function of the distance $||\mathbf{z}||_2$ from $\mathbf{x}$ for random and adversarial perturbations $\mathbf{z}$. (Right) Largest singular value of the linear operator $\mathbf{J}_f(\mathbf{x} + \mathbf{z})$ as a function of the magnitude $||\mathbf{z}||_2$ of random and adversarial perturbations $\mathbf{z}$. The dashed vertical line indicates the $\epsilon$ used during adversarial training. Curves were aggregated over 200 samples from the test set.

Figure 11: (Left) Classification accuracy as a function of perturbation strength $\epsilon$. (Right) Alignment of adversarial perturbations with dominant singular vector of $\mathbf{J}_f(\mathbf{x})$ as a function of perturbation magnitude $\epsilon$. The dashed vertical line indicates the $\epsilon$ used during adversarial training. Curves were aggregated over 2000 samples from the test set.

## 7.17 $\ell_\infty$-norm Constrained Projected Gradient Ascent

Additional results against $\ell_\infty$-norm constrained PGA attacks are provided in Figures 12 & 13. Note that all adversarial and regularized training methods are robustifying against $\ell_2$ PGA, or regularizing the spectral (2, 2)-operator norm, respectively. Results of adversarial training using $\ell_\infty$-norm constrained PGA and their equivalent regularization methods can be found in Section 7.18. The conclusions remain the same for all the experiments we conducted.

Figure 12: (Left) Deviation from linearity $||\phi^{L-1}(\mathbf{x} + \mathbf{z}) - (\phi^{L-1}(\mathbf{x}) + \mathbf{J}_{\phi^{L-1}}(\mathbf{x})\mathbf{z})||_2$ as a function of the distance $||\mathbf{z}||_2$ from $\mathbf{x}$ for random and $\ell_\infty$-PGA adversarial perturbations $\mathbf{z}$. (Right) Largest singular value of $\mathbf{J}_{\phi^{L-1}}(\mathbf{x} + \mathbf{z})$ as a function of the magnitude $||\mathbf{z}||_2$ of random and $\ell_\infty$-PGA adversarial perturbations $\mathbf{z}$. The dashed vertical line indicates the $\epsilon$ used during adversarial training. Curves were aggregated over 200 test samples.

Figure 13: (Left) Classification accuracy for an $\ell_2$-norm trained network on $\ell_\infty$-norm perturbations with $\epsilon$ (measured in 8-bit). (Right) Alignment of $\ell_\infty$-PGA adversarial perturbations with dominant singular vector of $\mathbf{J}_f(\mathbf{x})$ as a function of perturbation magnitude $\epsilon$. The dashed vertical line indicates the $\epsilon$ used during adversarial training. Curves were aggregated over 2000 samples from the test set.

## 7.18 Data-Dependent $\ell_\infty$-norm regularization

Figure 14 shows results against $\ell_\infty$-norm constrained PGD attacks when networks explicitly either use $\ell_\infty$-norm constrained adversarial training or, equivalently, regularize the $(\infty, 2)$-operator norm of the network. The conclusions remain the same for all the experiments we conducted.

Figure 14: (Left) Classification accuracy for an $\ell_\infty$-norm trained network on $\ell_\infty$-norm perturbations with $\epsilon$ (measured in 8-bit). (Right) Alignment of $\ell_\infty$-PGA adversarial perturbations with dominant singular vector of $\mathbf{J}_f(\mathbf{x})$ as a function of perturbation magnitude $\epsilon$. The dashed vertical line indicates the $\epsilon$ used during adversarial training. Curves were aggregated over 2000 samples from the test set.

## 7.19 SVHN

Figure 15 shows results against $\ell_2$-norm constrained PGD attacks on the SVHN dataset. As can be seen, the behavior is very comparable to our analogous experiment in the main section.

Figure 15: Classification accuracy on SVHN. Curves were aggregated over 2000 samples from the test set.

## Footnotes

[4]Note that for $p \in \{1, \infty\}$ the maximizer might not be unique, in which case we simply choose a specific representative.

[5]Note that technically equality only holds for $p, p^* \in (1, \infty)$. But one can easily check that the explicit expressions for $p \in \{1, \infty\}$ are in fact optimal. See comment after Equation 1.1 in [21].

[6] The proof is easier yet less elegant if one alternatively resorts to case distinctions.