[Reviews · NeurIPS 2020]

Review 1

Summary and Contributions: This paper first formulates a data-dependent form of operator norm regularization on the Jacobian of a feed-forward neural network. They then show that this regularization scheme is equivalent to the well-known adversarial training paradigm of Madry et al. They empirically show that both adversarial training and their regularization scheme shrink the singular values of the Jacobian of a trained model and that models trained using these schemes are significantly more linear around then data than are regularly trained models.

Strengths: + The connection between the well-known data- _independent_ spectral normalization scheme of Miyato et al. and the data _dependent_ scheme introduced here is both simple and compelling. + Given recent progress toward understanding the success of PGD and the prevalence of adversarial examples, this work is well-motivated. + Theorem 1 is quite compelling. I have seen many works that try to explain why PGD is one of the only defenses that holds up well against a variety of attacks. And this paper makes the best case for PGD that I have seen. + The experiments are very good. The authors made a clear effort to be very thorough here and it shows. A variety of experimental settings are considered. It may not be surprising that for example data-dependent SNR dampens the singular values of the Jacobian more effectively than the (admittedly looser bound in) data independent SNR. However, it was quite important to the fidelity of the claims made to verify this, and the authors do so successfully. - The discussion in the appendix about Frobenius norm regularization is necessary as it would be a natural question one might ask.

Weaknesses: - The theorem may slightly overstate its result in the following way: it seems that in order for this correspondence between adversarial training and the proposed regularization scheme to hold, \epsilon must be quite small. That is, we are assuming here that all of the points in an \epsilon ball around some data point x are mapped by the model to the same activation pattern \phi_x (i.e. that B_\epsilon^p(x) \subset X(\phi_x)). I would imagine that this may not hold for "realistic" values of \epsilon (e.g. 8/255) all the time. Indeed, my concern is that while this theorem is certainly compelling, it may be the case that it only holds for \epsilon so small that it may not hold in practice. Perhaps the authors can clarify here. I see there is an experiment to this effect in Section 7.16, but this seems to be for only one data point. [EDIT: post-rebuttal] Based on the authors response and a closer look at Section 5.4, I'm satisfied that the authors looked into this potential weakness and were able to add explanation as to its implications.] - Figure 1 is too small to really be useful. It's not really clear what the arrows represent. A more detailed and larger figure here would be appreciated. - The notation when describing the power iteration is a bit strange. This is a small thing, but I think that it would make more sense just to rearrange the steps. For example, in (6) it would be more clear to write \tilde{u} \gets ..., then u_k \gets ..., then \tilde{v}\gets ..., and finally v_k\gets ... so that you have these steps written in the order that you apply them.

Correctness: I looked through the proofs in the appendix and everything seems sound to me.

Clarity: This paper is very well written . The sections are clearly defined and the narrative flows well from one section the next. One typo I found: on page 3 near the bottom: totherther --> together

Relation to Prior Work: The related work is a little bit brief. A slightly more detailed related works section should probably be included in the final version.

Reproducibility: Yes

Additional Feedback: I really enjoyed this paper. The writing was very good, the ideas were compelling, and the contribution is impressive. A clear accept from my perspective.


Review 2

Summary and Contributions: This paper establishes a theoretical link between adversarial training and operator norm regularization. Specifically, this paper provides a data-dependent variant of spectrum norm regularization and proves that l_p norm constrained PGD with an l_q norm loss is equivalent to data-dependent (p,q) operator norm regularization. This reveals the connection between the network’s sensitivity to adversarial examples and its spectrum properties. Experiments support the theoretical findings.

Strengths: This paper presents a global spectral norm regularization for training robust models against adversarial examples. This paper shows that adversarial training (using l_q norm loss on the output logits) is a form of operator norm regularization, and confirms that a network’s sensitivity to adversarial examples is tied to its spectral properties. Experiments are conducted to support the theoretical findings.

Weaknesses: The theoretical results are not fascinating. Basically, this paper only shows the connection between adversarial training and data-dependent operator norm regularization, while we are still not clear about how such regularization affects the training of robust classifiers, and how it characterizes the sensitivity of the model to adversarial examples. The assumption made in this paper is not consistent with the practice. In particular, the derived theory requires that the loss function is l_q norm between the logits of the clean and perturbed inputs, while in practice people prefer cross-entropy or KL-divergence based loss functions. Additionally, Theorem requires extremely small epsilon such that B^p_epsilon(x)\subset X(phi_x) throughout the entire training period, which is also not practical.

Correctness: They are correct.

Clarity: The paper is clearly written and easy to follow.

Relation to Prior Work: yes

Reproducibility: Yes

Additional Feedback: Based on the experiment, the authors showed the local linearity and activation patterns of the neural network after training. Can you also plot the same figures regarding the model throughout the training, it is interesting to explore whether adversarial training/operator norm regularized training can help stabilize the activation pattern change against adversarial examples. ########## After reading the authors' response 1. I agree that It is also necessary to include a small amount of activation pattern changes in Theorem 1. For example, a better way of presenting Theorem 1 is to show the connection between adversarial training and operator norm regularization under the assumption that the number of activation pattern changes is upper bounded by some small quantity. What if we only consider the target logit? Then the Jacobian matrix will be a vector, will the theoretical result still hold? Can we observe the same singular value spectrum in the experiment? ########## after reading authors' rebuttal Thanks for pointing out related works that use lq norm losses for adversarial training. One thing I have to mention is that actually logit pair algorithms still use standard adversarial examples rather than generating them by maximizing lq norms. I agree that it is better to move the discussion of cross-entropy loss to the main part of the paper.


Review 3

Summary and Contributions: This paper builds the link between adversarial training and operator norm regularization for the learning by neural network and shows that $l_p$ norm constrained projected gradient ascent based adversarial training with an $l_q$ norm loss on the logit of clean and perturbed inputs is equivalent to (p,q) norm regularization. Empirically, experimental results verify the theoretical discussions. ** I have read all reviews and the rebuttal from the authors. After discussions, I believe that my evaluation is fair and proper. **

Strengths: This paper proposes a data-dependent spectral norm regularization variant which directly regularizes large singular values of a neural network. And by Theorem 1, it proves that $l_p$-norm constrained projected gradient ascent with an $l_q$-norm loss on the logits of clean and perturbed inputs is equivalent to data-dependent (p, q) operator norm regularization.

Weaknesses: This paper only discusses and proves the case for adversarial training with an $l_q$-norm loss. For other types of losses, it is not clear. The practical applicability appears to be narrow.

Correctness: In this paper, it uses the dataset CIFAR10 to verify the theoretical discussions and derivations. It seems insufficient to validate the effectiveness of an algorithm with only one data set. More empirical validations are needed.

Clarity: Basically it is clearly presented. In this paper, there are several typos or formatting issues needed to correct, such as the format consistency of conference references [26] and [27].

Relation to Prior Work: Yes. This paper discusses the differences with other prior work.

Reproducibility: Yes

Additional Feedback: More experiments on more datasets are needed to validate the effectiveness of the theoretic analysis in this paper.

[Author Response · NeurIPS 2020]

We would like to thank the reviewers for their positive and constructive feedback. We hope that with our detailed
answers below we can initiate a fruitful discussion that resolves all concerns.

**— R1.1): Validity of Theorem for large $\epsilon$. Experiment in Sec. 7.16.** R1 is correct, the Theorem establishes *exact*
equivalence only for $\epsilon$ small enough s.t. $\mathcal{B}_\epsilon^p(\mathbf{x}) \subset X(\phi_\mathbf{x})$. However, we experimentally verify that the correspondence
holds *approximately* (to a very good degree) in a region much larger than $X(\phi_\mathbf{x})$: see Sec. 5.4 "Validity of linear
approx.", specific. Fig. 4 (left), as well as Sec. 5.5 "Activation Patterns" specific. Fig. 5 and Fig. 9. Note, Fig. 9 shows the
*average* (incl. std. errors) over $\mathbf{x}$ and corresp. adv. $\mathbf{x}^*$ from the test set (not just a single data point). These experiments
confirm that $\mathbf{J}_f(\mathbf{x})$ is a good approx. (negligible deviation from linearity) and that the activation pattern change is small
($\sim 3\%$ in adv., $\sim 1\%$ in random directions) within $\mathcal{B}_{\epsilon^*}^p(\mathbf{x})$ for *realistic* $\epsilon^*$ commonly used during AT.

**Further comments:** We will clarify the power iteration notation, expand the related work and move the Frobenius
norm regularization to the additional ninth page for the camera-ready version. Thank you for your high-quality review.

**— R2.1): Still not clear how such regularization affects the training of robust classifiers and how it characterizes**
**the sensitivity of the model to adversarial examples.** Our Theorem confirms that a network's sensitivity to adversarial
examples is characterized through its spectral properties: it is the dominant singular vector (resp. the maximizer $\mathbf{v}^*$
in Lemma 1) corresponding to the largest singular value (resp. the $(p, q)$-operator norm) that determines the optimal
adversarial perturbation and hence the sensitivity of the model to adversarial examples. The effect of AT and d.d. ONR
on the training of robust classifiers is twofold: (i) they dampen the singular values, see Sec. 5.2 and specific. Fig. 2 and
(ii) they give rise to models that are significantly more linear around data than normally trained ones, see Sec. 5.4 and
specific. Fig. 4 (left) as well as Sec. 5.5 specific. Fig. 5 and Fig. 9. Note also that our results directly explain why input
gradient regularization and FGM based AT do not sufficiently protect against iterative adversarial attacks, see Sec. 5.2
and in particular Sec. 7.8 in the Appendix. We will add a summary at the end of the theory section to emphasize these
contributions more thoroughly.

**R2.2) $\ell_q$-norm loss between logits of the clean and perturbed inputs not consistent with practice.** It is not
uncommon to use $\ell_q$-norm losses for adversarial training. See for instance: [1] Harini, Kurakin, and Goodfellow,
"Adversarial logit pairing" or [2] Sabour et al., "Adversarial manipulation of deep representations" (both use an $\ell_2$-norm
loss on the logits / internal representations). We will add additional pointers to clarify this.

Note also that we have a Sec. 7.7 "Cross-Entropy based Adversarial Loss Function" in the Appendix, where we discuss
the effect of the loss function on the directional derivative in the power-method like formulation of AT. In particular, the
gradient of the loss w.r.t. the logits of the classifier takes the same "prediction - target" form for both the sum-of-squares
error as well as the softmax cross-entropy loss, see "A note on canonical link functions". We will move some of this
discussion to the main part in the camera-ready version to emphasize this connection.

**R2.3) Theorem requires small $\epsilon$.** See reply to **R1.1)**.

**R2.4) Plot local linearity and activation patterns during training. Explore whether AT / d.d. ONR can help**
**stabilize activation pattern changes against adversarial examples.** It is clear from Figs. 5 & 9 that AT and d.d. ONR
do improve the stability of activation patterns against adversarial examples, as both d.d. ONR and AT significantly
increase the size of the ReLU cells (in both random and adv. directions) compared to the normally trained model.
During training, we see a gradual progression towards the end-state. We will add the plots to the camera-ready version.

**R2.5) Generalize Theorem 1 to allow for activation pattern changes.** Proving such an extension for the "approxi-
mate correspondence" between AT and d.d. ONR is *highly non-trivial* and thus out-of-scope of the current paper: one
would have to take into account how much "nearby" Jacobians can change based on the crossings of ReLU boundaries,
which is complicated by the fact that the impact of such crossings depends heavily on (i) the specific activation pattern
at input $\mathbf{x}$, (ii) the precise values of the weights and biases in the network, and (iii) where in the network the units that
change their state are. We will add a note to highlight these challenges.

**R2.6) What if we only consider the target logit? Then the Jacobian is a vector, will the theoretical result still**
**hold? How will the spectrum look?** This can be formulated as a special case of our analysis and our results would
still hold (the optimal perturbation would align with the "Jacobian vector" and there would only be one singular value).
Note, however, that in general, the effectiveness of adversarial perturbations depends on the relative increase of one
logit over the decrease of another.

**— R3.1) Paper only proves correspondence for adversarial training with an $l_q$-norm loss.** See reply to **R2.2)**

**R3.2) Further datasets are needed to validate the effectiveness of the theoretical analysis.** We have confirmed
all our experiments on SVHN and TinyImageNet. The results and conclusions hold as expected. We will add the
corresponding plots to the camera-ready version.

**R3.3) Formatting issues, e.g. in bibliography.** We will standardize the formatting. Thank you for the pointer.

[Meta-Review · NeurIPS 2020]

This paper establishes a connection between adversarial training and operator norm regularization, which provides further insight into robustness in neural networks. The theoretical results are interesting and the analysis is thorough, including an empirical evaluation. This work should be interesting to anyone studying the theoretical properties and practical implications of adversarial training.